# The ion-ion recombination coefficient $\alpha$: Comparison of temperature- and pressure-dependent parameterisations for the troposphere and stratosphere

Marcel Zauner-Wieczorek[1], Joachim Curtius[1], Andreas Kürten[1]

[1]Institute for Atmospheric and Environmental Sciences, Goethe University Frankfurt am Main, Frankfurt am Main, 60629, Germany

*Correspondence to*: Marcel Zauner-Wieczorek (zauner-wieczorek@iau.uni-frankfurt.de) and Andreas Kürten (kuerten@iau.uni-frankfurt.de)

**Abstract.** Many different atmospheric, physical and chemical processes are affected by ions. An important sink for atmospheric ions is the reaction and mutual neutralisation of a positive and negative ion, also called ion-ion recombination. While the value for the ion-ion recombination coefficient $\alpha$ is well-known for standard conditions (namely $1.7 \cdot 10^{-6}$ cm$^3$ s$^{-1}$), it needs to be calculated for deviating temperature and pressure conditions, especially for applications at higher altitudes of the atmosphere. In this work, we review the history of theories and parameterisations of the ion-ion recombination coefficient,

focussing on the temperature and pressure dependencies and on the altitude range of between 0 and 50 km. Commencing with theories based on J. J. Thomson's work, we describe important semi-empirical adjustments as well as field, model and laboratory data sets, followed by short reviews of binary recombination theories, model simulations, and the application of ion-aerosol theories to ion-ion recombination. We present a comparison between theories, parameterisations, field, model, and laboratory data sets to conclude on favourable parameterisations. While many theories agree well with field data above

approximately 10 km altitude, the nature of the recombination coefficient is still widely unknown between Earth's surface and an altitude of 10 km. According to the current state of knowledge, it appears reasonable to assume an almost constant value for the recombination coefficient for this region, while it is necessary to use values that are adjusted for pressure and temperature for altitudes above 10 km. Suitable parameterisations for different altitude ranges are presented and the need for future research, be it in the laboratory or by means of modelling, is identified.

# 1 Introduction

Earth's atmosphere is not only a neutral mixture of gases, but also contains gas-phase ions that are crucial to the phenomena of atmospheric electricity. They play a central role in meteorological processes in thunderstorms (Sagalyn et al., 1985), maintaining the global atmospheric electrical circuit (Harrison, 2004), the formation of aerosol particles with the ion-induced nucleation mechanism (Hirsikko et al., 2011), and the propagation of radio waves in the ionosphere (Basu et al., 1985), to

name but a few processes. It is, therefore, important to understand the production and loss of atmospheric ions. There are several sources of ions in the atmosphere, of which ionisation by galactic cosmic rays (GCRs) is the most important (Bazilevskaya et al., 2008). Close to the ground, ionisation by the radioactive decay of radon as well as lightning are additional sources of atmospheric ions (Viggiano and Arnold, 1995). Further, minor sources of ionisation in the atmosphere include solar cosmic rays (also called solar energetic particles, SEPs) and magnetospheric electrons (Bazilevskaya et al., 2008). The two

important sinks for atmospheric ions are the reaction of a positive ion and negative ion, the so-called recombination, as well as the condensation of ions onto aerosol particles (Viggiano and Arnold, 1995). The ion-ion recombination coefficient $\alpha$ describes the reaction rate of the recombination of a positive and negative ion in the gas phase; its unit is cm$^3$ s$^{-1}$, which is used throughout this work unless noted otherwise. There are two important recombination mechanisms: binary recombination, in which two ions of opposite sign recombine upon collision, and ternary recombination, in which one of the ions first collides

with a neutral gas molecule, i.e. the third body, to dissipate energy in order to recombine successfully with the second ion. The latter process is, therefore, also called three-body trapping. When both the binary and ternary processes are included in a theory

or parameterisation, it is called total recombination. While ion-ion recombination concerns the recombination of atomic or molecular ions or small molecular ion clusters, ion-aerosol attachment regards the interaction between an ion and a charged or neutral aerosol particle. Typically, aerosol particles are defined to have a size of 1 nm or bigger. As the ion-aerosol attachment coefficient depends on the size of the aerosol particle, the ion-ion recombination coefficient can be viewed as a special case of the former if the "aerosol particle" is considered to have ionic size and is singly charged.

In this work, we focus on ion-ion recombination and summarise the history and fundamentals of the theory behind it in Sect. 2, followed by a description of the theories of the binary ion-ion recombination process in Sect. 3. In Sect. 4 we discuss the field and laboratory measurements and subsequent semi-empirical parameterisations of ion-ion recombination. We focus on the applicability of the theory to atmospheric conditions, especially for the troposphere and stratosphere, i.e. in an altitude range of 0 to 50 km. In Sect. 5 we describe the application of ion-aerosol theories to the ion-ion recombination, followed by an overview of numerical simulations in Sect. 6. Later, we compare the available parameterisations and theories with field data, laboratory data and a model simulation for the atmospheric altitude range of 0 to 50 km in Sect. 7. The determination of the three-body trapping sphere radius and the collision probability in the limiting sphere (a concept used in different theories) can be found in Sect. 8. Finally, we conclude on the applicability of the discussed theories to atmospheric conditions and identify the demand for future research in Sect. 9. To improve the readability, we stick to the conventionally used units of hPa and atm for pressure, $cm^3 s^{-1}$ for the recombination rate and related quantities, eV for the electron affinity, and km for the atmospheric altitude; otherwise, we use the SI units.

## 2 The fundamental theories

The theoretical foundation of the recombination of gaseous ions was laid down by J. J. Thomson and Ernest Rutherford. The theory based on their approach is referred to as Thomson theory in the literature. In their work "On the Passage of Electricity of Gases exposed to Röntgen Rays", Thomson and Rutherford (1896) discuss, for the first time, the sources and sinks of ions in the gas phase. In their experimental setup, the source of ions are X-rays while the sinks are the recombination of negative and positive ions as well as losses to the electrodes. They describe the temporal change of the number concentration of ions $n$ according to Eq. (1):

$$\frac{dn}{dt} = q - \alpha n^2 - L ,\tag{1}$$

where $t$ is the time, $q$ is the ion production rate, and $L$ is the loss rate to the electrodes. This formula already includes the assumption that the number concentrations of negative and positive ions, $n_-$ and $n_+$, are about equal and, therefore, the product $n_- n_+$ can be simplified to $n^2$. They conclude that when the electrode current is small ($L \approx 0$) and the system is in a steady state ($dn/dt = 0$), the number concentration of gas-phase ions can be calculated in a simple way (Eq. (2)):

$$n^2 = \frac{q}{\alpha}.\tag{2}$$

As per today's convention, $q$ is the production rate for ion pairs so that $n$ in Eq. (1) and (2) must be specified to be either $n_+$ or $n_-$, not to be confused with $n_{total} = n_+ + n_-$. Equation (2) can be rearranged to determine $\alpha$ when the ion pair production rate and the number concentration of positive or negative ions are known. A few years later, in 1906, the Nobel Prize in Physics was

awarded to J. J. Thomson for his studies on the electrical conductivity of gases.

Soon after Thomson and Rutherford's publication, several experiments to determine the ion-ion recombination coefficient were pursued by different scientists. It was shown that $\alpha$ is dependent on the chemical composition of the surrounding gas, as well as on the temperature and pressure. Here, we focus on experiments in air. Many of these first approaches have been reviewed by Lenz (1932), who himself had developed a sophisticated experimental setup in order to control losses due to

diffusion and deposition on walls. It is remarkable that even during this time, the determined values of $\alpha$ are similar to the one used today and have not changed significantly since then. For standard conditions, i.e. 273.15 K and 1013 hPa, Thirkill (1913) determined a value of $1.7 \cdot 10^{-6}$ cm$^3$ s$^{-1}$, while Thomson (1924) determined a value of $2.0 \cdot 10^{-6}$ cm$^3$ s$^{-1}$. Lenz (1932) reported $(1.7 \pm 0.1) \cdot 10^{-6}$ cm$^3$ s$^{-1}$ for the conditions of 291.15 K and 1013 hPa. The value for $\alpha$ used nowadays is $1.6 \cdot 10^{-6}$ cm$^3$ s$^{-1}$ (e.g. Franchin et al., 2015) and is taken from Israël (1971) (which is the English translation of the first edition in German: Israël

(1957)). In addition, Gardner (1938) reported the value of $2.1 \cdot 10^{-6}$ cm$^3$ s$^{-1}$ for pure oxygen, 1013 hPa and 298.15 K. Sayers (1938) reported a value of $2.3 \cdot 10^{-6}$ cm$^3$ s$^{-1}$, while Nolan (1943), who has also reviewed previous works, concluded on $1.4 \cdot 10^{-6}$ cm$^3$ s$^{-1}$. Within a particular uncertainty range, these values do agree quite well and no further ado appears to be necessary to discuss this value. However, the values for $\alpha$ differ tremendously when temperatures are lower than 273.15 K and pressures are lower than 1013 hPa, as Lenz (1932) has already shown. Loeb (1960) pointed out that measurement techniques

were not sophisticated enough and gases not pure enough before the 1950s to be able to determine accurately the ion-ion recombination. In any case, a correct value for $\alpha$ is crucial for the analysis of field data and for the calculations of atmospheric models at higher altitudes in the atmosphere where the temperatures and pressures are different from those at ground level. This calls for a good understanding of the mechanisms involved in ion-ion recombination and a solid parameterisation of $\alpha$.

In a later work, Thomson (1924) explains his theory in more detail and provides a kinetic derivation of the recombination

coefficient. In his approach, recombination occurs when the two oppositely charged ions each collide with a neutral molecule of the surrounding gas within a certain sphere $d_T$ around the respective ions. It is defined as the sphere in which the ions of opposite signs experience Coulomb attraction (Loeb, 1960), thus, it can be derived from equalising the Coulomb potential energy, $e^2 (4\pi \, \varepsilon_0 \, d_T)^{-1}$, and the thermal energy of motion of the surrounding molecules and ions in the absence of an electrical field, $1.5 \, k_B T$ (Loeb, 1960), as shown in Eq. (3):

$$d_T = \frac{e^2}{4\pi \cdot \varepsilon_0 \cdot 1.5 \, k_B T},$$    (3)

where $e$ is the elementary charge, $\varepsilon_0$ is the vacuum permittivity, $k_B$ is the Boltzmann constant, and $T$ is the temperature. Loeb (1960) stresses that the pre-factor value of 1.5 for the thermal energy is debated, ranging from 1 (Tamadate et al., 2020b), 1.5 (Thomson, 1924), and 2.4 (Natanson, 1959) to 6 (Loeb and Marshall, 1929), amongst others. Loeb and Marshall (1929)

approximate the radius $d_T$ to be in the order of 10 nm when the value of $6 k_B T$ is used for the thermal energy. Thus, a rough
estimate of 10 to 60 nm for $d_T$ can be derived from different Thomsonian theories.

Thomson deduced that $\alpha$ is dependent on the average speeds of the positive and negative ion, $v_+$ and $v_-$, respectively, according to Eq. (4) for low pressures and Eq. (5) for high pressures:

$$\alpha = 2\pi \cdot (v_+{}^2 + v_-{}^2)^{0.5} \cdot d_T{}^3 \cdot \left(\frac{1}{\lambda_+} + \frac{1}{\lambda_-}\right), \tag{4}$$

where the pressure is low, i.e. $d_T \lambda_{ion}{}^{-1}$ is small and $\lambda_+$ and $\lambda_-$ are the mean free paths of the positive and negative ions,
respectively, and $\lambda_{ion}$ is the mean free path of one ion.

$$\alpha = 2\pi \cdot (v_+{}^2 + v_-{}^2)^{0.5} \cdot d_T{}^2, \tag{5}$$

where the pressure is high, i.e. $d_T \lambda_{ion}{}^{-1}$ is large. From these equations, Thomson (1924) deduced that the recombination coefficient is proportional to the pressure for low pressures (because of the sum of the reciprocal mean free paths of the ions), whereas it is independent of the pressure for high pressures. This was supported by the measurements of Thirkill (1913) who
found $\alpha$ to be proportional to the pressure throughout the measurement range of approximately 200 to 1000 hPa. Thus, the pressure regime of 1013 hPa and below is included in the low-pressure scenario. The transition pressure from the low-pressure to the high-pressure regime is clearly above 1013 hPa and, thus, beyond the concern of atmospheric application. The temperature dependence is given as $\alpha \sim T^{-2.5}$ for low pressures and $\alpha \sim T^{-1.5}$ for high pressures because $d \sim T^{-1}$ and $v_{+,-} \sim T^{0.5}$ (Thomson, 1924). Hence, the recombination coefficient decreases with rising temperature for pressures below 1013 hPa. For
the troposphere, this leads to a somewhat counterbalancing effect on $\alpha$ for increasing altitudes when both the temperature and the pressure drop simultaneously.

Another approach to explain the recombination of ions was introduced by Langevin (1903a, 1903b) whose ansatz is based on the speeds of ions in an electrical field, as opposed to the later thermodynamic approach of Thomson. To account for the effectiveness of collisions of a negative and positive ion with regard to recombination, Langevin introduced the ratio of
successful recombinations per collision, $\varepsilon_L$, which is included in the formula proposed by him to determine $\alpha$ (Eq. (6)); this probability is of empirical nature and was not further defined by a formula.

$$\alpha = \frac{e}{\varepsilon_0} \cdot (\mu_+ + \mu_-) \cdot \varepsilon_L, \tag{6}$$

where $\mu_+$ and $\mu_-$ are the ion mobilities of the positive and negative ions, respectively, defined by Eq. (7a) and (7b):

$$v_+ = \mu_+ \cdot \left(E + \frac{e}{4\pi \cdot \varepsilon_0 \cdot r^2}\right), \tag{7a}$$

$$v_- = -\mu_- \cdot \left(E + \frac{e}{4\pi \cdot \varepsilon_0 \cdot r^2}\right), \tag{7b}$$

where $E$ is the external electrical field and $e\,(4\pi\,\varepsilon_0\,r^2)^{-1}$ is the electrical field produced by the ions. Here, $r$ is the distance between the two ions of opposite charge (Langevin, 1903a). The consideration of the recombination efficiency leads to an adapted term for the recombination sink (Langevin, 1903a), shown in Eq. (8):

$$\frac{\mathrm{d}n_\pm}{\mathrm{d}t} = -\frac{e}{\varepsilon_0} \cdot (\mu_+ + \mu_-) \cdot \varepsilon_L \cdot n_+ \cdot n_-. \tag{8}$$

Langevin (1903b) further determined the pressure ($p$) dependence of $\varepsilon_L$ (and, thus, of $\alpha$). For 1013 hPa, $\varepsilon_L = 0.27$ and $\varepsilon_L \sim p^2$ (and, thus, $\alpha \sim p^2$) for pressures below 1013 hPa. This, however, is in contradiction to Thomson (1924) who stated that $\alpha \sim p$ for low pressures. Loeb and his colleagues later argued that the assumptions made by Langevin to calculate $\varepsilon_L$ are based on incorrect, sometimes even antithetical assumptions. They even stated that this correction factor was only introduced to make the experimental results fit the theoretical ones. The application of Langevin theory is only considered valid for very high

pressures (above approximately 10 atm) (Loeb and Marshall, 1929; Gardner, 1938; Loeb, 1960) and is, thus, not within the focus of this work.

Loeb and Marshall (1929) further advanced and refined the Thomson theory; they introduced a probability term, similar to Langevin, for collisions leading to recombination, extending Eq. (5) to a more refined Eq. (9):

$$\alpha = \pi \cdot d_T^2 \cdot (v_+^2 + v_-^2)^{0.5} \cdot \left[ 1 - \frac{\lambda_{ion}^2}{2d_T^2} \cdot \left( 1 - e^{-2d_T/\lambda_{ion}} \cdot \left( \frac{2d_T}{\lambda_{ion}} + 1 \right) \right) \right]^2. \tag{9}$$

In subsequent works, the ratio of the doubled collision sphere radius and the mean free path of the ion, $2d_T \cdot \lambda_{ion}^{-1}$, is often denoted as $x$. With a number of assumptions and simplifications, and together with validation from experimental work, Gardner (1938) summarised the previous findings and advanced them to a set of equations (Eq. (10) to (13)) that contain macroscopic quantities that are more accessible for direct observation:

$$\alpha = 1.9 \cdot 10^{-5} \cdot \left( \frac{T_0}{T} \right)^{1.5} \cdot \left( \frac{1}{m_{ion}} \right)^{0.5} \cdot \varepsilon_T, \text{ where} \tag{10}$$

$$\varepsilon_T = 2w_T - w_T^2, \tag{11}$$

$$w_T = 1 - 2 \cdot \frac{[1 - e^{-x\prime} \cdot (x\prime + 1)]}{x\prime^2}, \text{ and} \tag{12}$$

$$x' = 0.810 \cdot \left( \frac{T_0}{T} \right)^2 \cdot \left( \frac{p}{p_0} \right) \cdot \frac{\lambda_{air}}{\lambda_{ion}}, \tag{13}$$

where $T_0$ is the temperature at standard conditions (i.e. 273.15 K), $m_{ion}$ is the ion mass in Da, $\varepsilon_T$ is the recombination probability upon collision, $p_0$ is the pressure at standard conditions (i.e. 1013.25 hPa), $\lambda_{air}$ is the mean free path of the surrounding air, and

the ratio $\lambda_{air} \cdot \lambda_{ion}^{-1} = 5$. Here, $x$ is not defined by the ratio of $d_T$ and $\lambda_{ion}$ anymore, but as a function of $T$, $p$, and the ratio $\lambda_{air} \cdot \lambda_{ion}^{-1}$, and is, therefore, denoted as $x'$. Importantly, Loeb and Marshall (1929) also discuss the limitations of their approach. Firstly, the exact masses of the ions are unknown, as clustered ions and ions from impurities in the sample gas can also occur. This has a non-negligible effect on the value of the recombination coefficient. They argue that this circumstance could be the reason for the variation of results between different authors. Apart from the difference in ion mass, the property of free electron pairs

in the surrounding gas may also influence the recombination (Loeb and Marshall, 1929). Secondly, based on the observation that at low temperatures, the measured $\alpha$ value is much smaller than the calculated one, they discuss whether the presumed power of $-1.5$ for the $T$ dependence might be inaccurate.

A detailed overview of the different theories and their experimental validations can be found in Loeb (1960) (second edition of Loeb (1955)) where he discusses ion-electron and ion-ion recombination. For the latter, cases of $\alpha$ particle- or X-ray-induced

ion production are also described that feature a non-uniform spatial ion distribution. Until the beginning of the 1980s, it was hypothesised that, in general, ions are not uniformly distributed in the atmosphere, because the ions are produced along the GCR paths and diffusion may not be sufficiently fast. Bates (1982) showed that ions are mixed sufficiently rapidly in the atmosphere so that the assumption of a uniform ion concentration of the "volume recombination" theories, described in the following, is valid. In Table 1, a selection of recombination theories discussed in detail by Loeb is given. Above atmospheric

pressure, Langevin theory is applied. Loeb subclassifies it, firstly, to the range of 20 to 100 atm where there is no diffusional approach of the ions towards each other because they are already within the Coulomb attractive radius $d_T$ and, secondly, to the range of 2 to 20 atm (called Langevin-Harper theory), where the initial distance of the ions $r_0$ is greater than $d_T$ and so they first have to diffuse towards each other. The subsequent collision inside $d_T$ is almost certain because of the high pressure. For the pressure range of 0.01 to 1013 hPa, i.e. for the lower and middle atmosphere, Thomson theory is applicable. Here, the

initial distance of the ions is greater than $d_T$ and the mean free path $\lambda_{ion}$, therefore a random diffusive approach is necessary. Within $d_T$, the collision probability $\varepsilon_T$ is less than 1. Below 0.01 hPa, i.e. in the ionosphere, the collision probability becomes almost 0 and, thus, the collision is then governed by the collision cross section. For super-atmospheric pressures (i.e. in the Langevin regime), $\alpha$ is dependent on $p^{-1}$ and proportional to $T$. In the regime where the Thomson theory should be applicable (i.e. from 0.01 to 1013 hPa), $\alpha$ is dependent $T^{-1.5}$, The pressure dependence of $\alpha$ is different in various Thomsonian theories;

while Thomson (1924) stated a proportional dependence (see Eq. (4)), it varies in the parameterisations of Gardner (1938), Israël (1957), and Loeb (1960) (see Eq. (10), (14), (15), respectively, with Eq. (11) to (13)): for approximately 500 to 1000 hPa, $\alpha$ is dependent on $p^{0.5}$ and below 500 hPa, it approaches $p^1$. In the cross section regime (i.e. < 0.01 hPa), $\alpha$ is independent of the pressure and dependent on $T^{-0.5}$.

**Table 1: Selection of ion-ion recombination theories described in detail by Loeb (1960). $d_T$ is the radius of mutual Coulomb attraction**
**between the two ions of opposite charge, $r_0$ is the initial distance of the two ions, and $\lambda_{ion}$ is the mean free path of one ion.**

| Theory | Pressure range | Conditions | $p$ and $T$ dependence | Description |
|---|---|---|---|---|
| Langevin | 20 to 100 atm | $d_T > r_0 > \lambda_{ion}$ | $p^{-1}$, $T^1$ | both ions inside $d_T$, no diffusive approach |
| Langevin-Harper | 1 to 20 atm | $r_0 > d_T > \lambda_{ion}$ | $p^{-1}$, $T^1$ | diffusion towards $d_T$, collision inside $d_T$ certain |
| Thomson | 0.01 to 1013 hPa | $r_0 > d_T \approx \lambda_{ion}$ | $p^{0.5...1}$, $T^{-1.5}$ | random diffusive approach, finite collision probability $\varepsilon_T$ |
| Collision cross section | < 0.01 hPa | $\lambda_{ion} > r_0 > d_T$ | no $p$ dep., $T^{-0.5}$ | collision probability $\varepsilon_T \approx 0$, collision driven by cross section (ionosphere) |

A detailed derivation of all theories and the above mentioned equations is given within Loeb (1960). In his work, the only variation in the Thomson parameterisation for $\alpha$ from the one presented by Gardner (1938) is the first factor in the formula for the recombination coefficient, as shown in Eq. (14):

$\qquad \alpha = 1.73 \cdot 10^{-5} \cdot \left(\frac{273}{T}\right)^{1.5} \cdot \left(\frac{1}{m_{\mathrm{ion}}}\right)^{0.5} \cdot \varepsilon_{\mathrm{T}}.$ $\hfill$ (14)

Israël (1957) has further altered this parameterisation by including the few experimental data available at that time into his parameterisation. In the derivation of the formula he used the value of $1.6 \cdot 10^{-6}$ cm$^3$ s$^{-1}$ for $\alpha$ for "normal conditions", however, he neither included a reference for this nor specified the normal conditions. These were probably 273.15 K and 1013.25 hPa. Furthermore, he stated that the recombination of small negative and small positive ions are accompanied by the recombination

of small and big ions and, also, of small ions with neutrals, so that a whole equation system of recombination rates would result. He proposed the slightly altered parameterisation of the small ion recombination that no longer includes the ion mass according to Eq. (15):

$\qquad \alpha = 1.95 \cdot 10^{-6} \cdot \left(\frac{273}{T}\right)^{1.5} \cdot \varepsilon_{\mathrm{T}}.$ $\hfill$ (15)

Contrary to previous authors, Israël used the value $\lambda_{\mathrm{air}} \cdot \lambda_{\mathrm{ion}}^{-1} \approx 3$ for air; Gardner (1938) and Loeb (1960) used the value of 5.

Note that in Israël's work, there is a typing error in the formula of $\varepsilon_{\mathrm{T}}$: Instead of $\varepsilon_{\mathrm{T}} = 1 - \frac{4}{x'^4} \cdot [1 - e^{-x'} \cdot (x' + 1)]^2$, the fraction in front of the brackets was erroneously given as $\frac{4}{x'^2}$.

Natanson (1959a) (English translation of the original in Russian: Natanson (1959b)) developed a theory to unify Thomson's (low pressure) and Langevin's (high pressure) approaches; this formula is given in Eq. (16), assuming two ions of identical mass:

$\qquad \alpha = \frac{\pi \cdot d_{\mathrm{N}}^2 \cdot v_{\mathrm{rel}} \cdot \varepsilon_{\mathrm{N}} \cdot \left[1 + \frac{e^2 \cdot \lambda}{4\pi \cdot \varepsilon_0 \cdot d_{\mathrm{N}} \cdot (d_{\mathrm{N}} + \lambda) \cdot k_{\mathrm{B}} T}\right] \cdot \exp\left(\frac{e^2}{4\pi \cdot \varepsilon_0 \cdot (d_{\mathrm{N}} + \lambda) \cdot k_{\mathrm{B}} T}\right)}{1 + \frac{\pi \cdot \varepsilon_0 \cdot d_{\mathrm{N}}^2 \cdot v_{\mathrm{rel}} \cdot \varepsilon_{\mathrm{N}} \cdot k_{\mathrm{B}} T}{e^2 \cdot D} \cdot \left[1 + \frac{e^2 \cdot \lambda}{4\pi \cdot \varepsilon_0 \cdot d_{\mathrm{N}} \cdot (d_{\mathrm{N}} + \lambda) \cdot k_{\mathrm{B}} T}\right] \cdot \left[\exp\left(\frac{e^2}{4\pi \cdot \varepsilon_0 \cdot (d_{\mathrm{N}} + \lambda) \cdot k_{\mathrm{B}} T}\right) - 1\right]},$ with $\hfill$ (16)

$\qquad d_{\mathrm{N}} = \frac{\lambda}{2} \cdot \left[\sqrt{1 + \frac{5e^2}{12\pi \cdot \varepsilon_0 \cdot k_{\mathrm{B}} T \cdot \lambda}} - 1\right],$ $\hfill$ (17)

$\qquad v_{\mathrm{rel}} = \sqrt{\frac{8 \cdot k_{\mathrm{B}} T}{\pi \cdot m_{\mathrm{red}}}},$ $\hfill$ (18)

$\qquad \varepsilon_{\mathrm{N}} = 2 w_{\mathrm{N}} - w_{\mathrm{N}}^2,$ $\hfill$ (19)

$\qquad w_{\mathrm{N}} = 1 - \frac{2}{x_{\mathrm{N}}^2} \cdot [1 - e^{-x_{\mathrm{N}}} \cdot (x_{\mathrm{N}} + 1)],$ and $\hfill$ (20)

$\qquad x_{\mathrm{N}} = \frac{2 \cdot d_{\mathrm{N}}}{\lambda},$ $\hfill$ (21)

where $d_{\mathrm{N}}$ is the ion-ion trapping distance, $v_{\mathrm{rel}}$ is the mean relative thermal speed of the ions, $\varepsilon_{\mathrm{N}}$ is the probability that one ion collides with a gas molecule while the other ion is at a distance $< d_{\mathrm{N}}$ (also named "absorption coefficient"), $D$ is the diffusion coefficient, and $m_{\mathrm{red}}$ is the reduced mass in kg. Note the use of $x_{\mathrm{N}}$ that depends on Natanson's $d_{\mathrm{N}}$ in Eq. (20), which is otherwise identical to Eq. (12). $D$ is the sum of $D_+$ and $D_-$, the diffusion coefficients of the positive and negative ion, respectively.

Tamadate et al. (2020b) suggested to exchange $v_{\text{rel}} \cdot D^{-1}$ by the reciprocal of the ion-ion mean free path, $\lambda^{-1}$, in the first fraction of the denominator in Eq. (16), based on the definition of $\lambda$ in Eq. (22):

$$\lambda = D \cdot v_{\text{rel}}^{-1} = (D_+ + D_-) \cdot \left(\frac{\pi \cdot m_{\text{red}}}{8 \cdot k_B T}\right)^{0.5}, \tag{22}$$

where $D_+$ and $D_-$ are calculated according to Eq. (23a) and (23b), respectively:

$$D_+ = D_{+,0} \cdot \frac{p_0}{p} \cdot \left(\frac{T}{T_0}\right)^{1.75} \text{ and} \tag{23a}$$

$$D_- = D_{-,0} \cdot \frac{p_0}{p} \cdot \left(\frac{T}{T_0}\right)^{1.75}, \text{ with} \tag{23b}$$

$$D_{+/-,0} = \frac{\mu_0 \cdot k_B T_0}{e}, \tag{24}$$

where $D_{+,0}$ and $D_{-,0}$ are the reference diffusivities calculated from the reference ion mobility at standard pressure ($p_0 = 1013.25$ hPa) and standard temperature ($T_0 = 273.15$ K), $\mu_0$, given by López-Yglesias and Flagan (2013) as $\mu_0 = 1.35 \cdot 10^{-4}$ m$^2$ V$^{-1}$ s$^{-1}$ for the ion mass of 90 Da. The temperature dependence of 1.75 for $D_{+,-}$ is taken from Tang et al.
(2014). Note that López-Yglesias and Flagan (2013) use $T^2$ and the Chapman-Enskog theory predicts $T^{1.5}$ (Chapman and Cowling, 1960).

In the course of time, additional sinks for atmospheric ions, other than the ion-ion recombination process, have been discussed. Lenz (1932) explained the strong deviations observed between several experimentally derived values for $\alpha$ by the authors' negligence of losses due to their experimental setups, for example, by wall losses. In addition, the attachment of ions to aerosol
particles suspended in the surrounding gas has been found to cause problems in the quantification of $\alpha$, especially in field studies performed in the atmosphere (Rosen and Hofmann, 1981; Morita, 1983), while Franchin et al. (2015), who conducted chamber experiments, included the aerosol sink and wall losses to their calculations. Furthermore, one has to bear in mind that the capabilities of the instruments and the purity of the gases were less advanced before the 1950s (Loeb, 1960). Therefore, results obtained before that time need to be considered with caution. Nevertheless, the theory of ion-ion recombination
experienced more advances in the following decades as discussed in the next sections.

## 3 Binary ion-ion recombination

In the previous section, the theories and parameterisations that concern the total ion-ion recombination, i.e. the combination of binary and ternary processes, were discussed. Commencing in the late 1970s, several groups examined the binary and ternary processes, respectively, in more detail. Hickman (1979) developed an approach to determine the binary recombination
coefficient, $\alpha_2$. Based on a complex potential model, the neutralisation of two ions of opposite sign is determined by an electron transfer from the negative to the positive ion. While the two ions approach one other, the electron transfer can occur when the potential curve of the initial state crosses at least one of the potential curves of the final states. In the semi-empirical Eq. (25), $\alpha_2$ depends on the temperature $T$, the reduced mass $m_{\text{red}}$, and the electron affinity $EA$ of the negative ion, i.e. its electron detachment energy:

$\quad \alpha_2 = 5.35 \cdot 10^{-7} \cdot \left(\frac{T}{300}\right)^{-0.5} \cdot m_{\text{red}}^{-0.5} \cdot EA^{-0.4},$ $\hfill$ (25)

where $m_{\text{red}}$ is in Da and $EA$ is in eV. Due to the mass and electron affinity dependencies, the recombination coefficient can vary by one order of magnitude or more, e.g. from $(49 \pm 20) \cdot 10^{-8}$ cm$^3$ s$^{-1}$ for NO$^+$ + O$^-$ to $(4.1 \pm 0.4) \cdot 10^{-8}$ cm$^3$ s$^{-1}$ for CClF$_2^+$ + Cl$^-$ (Hickman, 1979). The temperature dependence is $T^{-0.5}$; the mass dependence of $m_{\text{red}}^{-0.5}$ is in accordance to Gardner (1938) and Loeb (1960). The dependence on the electron affinity is unique compared to the other approaches.

$\quad$ Several experiments were performed to test this deduction. A recent approach was reported by Miller et al. (2012) who used the variable electron and neutral density attachment mass spectrometry (VENDAMS) method, utilising a flowing-afterglow Langmuir-probe (FALP) apparatus. This method is limited to atomic cations from noble gases. Miller et al. (2012) determined the rate coefficients of neutralisation reactions of several anions, among them SF$_{4-6}^-$, NO$_3^-$, and Br$_2^-$, with Ar$^+$ and Kr$^+$ at conditions of 300–550 K and a helium buffer gas number concentration of typically $3.2 \cdot 10^{16}$ cm$^{-3}$ (i.e. 1.3 hPa at 300 K).

$\quad$ They also summarised previous works. The resulting binary ion-ion recombination coefficients were found to be in the range of 2.5 to $5.6 \cdot 10^{-8}$ cm$^3$ s$^{-1}$ at 300 K, showing decreasing values for higher temperatures, with a typical uncertainty of $5 \cdot 10^{-9}$ cm$^3$ s$^{-1}$ (Miller et al., 2012). Shuman et al. (2014b) later pointed out that the rate coefficients involving Ar$^+$ should be uniformly increased by $4 \cdot 10^{-9}$ cm$^3$ s$^{-1}$. Miller et al. (2012) fitted the data to the parameterisation developed by Hickman (1979), resulting in adapted exponents for $T$, $m_{\text{red}}$, and $EA$, as shown in Eq. (26) and (27):

$\quad \alpha_2 = (3.2 \pm 1.4) \cdot 10^{-8} \cdot \left(\frac{T}{300}\right)^{-1.1 \pm 0.2} \cdot m_{\text{red}}^{-0.01 \pm 0.09} \cdot EA^{-0.04 \pm 0.23}$ for diatomic anions and $\hfill$ (26)

$\quad \alpha_2 = (2.8 \pm 1.0) \cdot 10^{-7} \cdot \left(\frac{T}{300}\right)^{-0.9 \pm 0.2} \cdot m_{\text{red}}^{-0.5 \pm 0.1} \cdot EA^{-0.13 \pm 0.04}$ for polyatomic anions. $\hfill$ (27)

$\quad$ Later, the mutual neutralisation reactions of di- and polyatomic cations with the halide anions Cl$^-$, I$^-$, and Br$^-$ were also studied (Shuman et al., 2014a). The cations were produced by transferring the charge from Ar$^+$ to neutral species such as O$_2$, NO or CF$_4$. It was found that the binary ion-ion recombination coefficients are primarily governed by the chemical nature of the

system (i.e. the locations of the curve crossings) for systems with two monoatomic ions recombining, while the physical nature of the system (e.g. the relative velocity of the ions) becomes dominant for systems with more than 4 or 5 atoms. For the latter, a good description of the rate constants is given by $2.7 \cdot 10^{-7} m_{\text{red}}^{-0.5} (T/300)^{-0.9}$ (Shuman et al., 2014a), thus, agreeing with the previous findings given in Eq. (27). In addition, experiments with heavier molecular ions such as C$_{10}$H$_8^+$, WF$_5^+$, and C$_6$F$_{11}^+$ support these findings (Wiens et al., 2015).

$\quad$ While most research in the field has been carried out on the recombination process itself, some works have also been devoted to study the product formation. For instance, Shuman et al. (2010) investigated the different product channels of the recombination of SF$_{4-6}^-$ with Ar$^+$. Besides simple electron transfer reactions, the elimination of F atoms was also observed. Subsequently, further parameterisations of the total ion-ion recombination coefficient based on laboratory experiments and field measurements in the troposphere and stratosphere were developed, as will be discussed in the next section.

## 4 Field and laboratory measurements and semi-empirical parameterisations

In the late 1970s, Smith and Church (1977) investigated the different influences of binary and ternary collisions on the recombination coefficient. They determined the recombination rates of $NO^+$ and $NO_2^-$ in an experimental setup for different temperatures and pressures typical for the atmosphere and inferred the binary ($\alpha_2$) and ternary ($\alpha_3$) recombination rates for different altitudes using helium as the carrier gas. They found that above 30 km, where air is less dense, the binary recombination is dominant, while below 30 km, where air is denser and three-body collisions become more likely, the ternary recombination is more important. For ground level, they determined a rather high value of $\alpha_3 = 3 \cdot 10^{-6}$ $cm^3$ $s^{-1}$. Interestingly, the total ion-ion recombination rate is almost constant throughout the whole troposphere according to their work. Only above 10 km, does the value decrease until an altitude of about 50 km. With regard to the temperature, they determined a dependency of $T^{-0.4}$ from their data for the binary recombination. For the ternary recombination, they expected a dependency of $T^{-2.5}$ to $T^{-3}$, while Fisk et al. (1967) even determined $T^{-4.1}$ in a recombination experiment with $Tl_2I^+$ and $TlI_2^-$. Smith and Church (1977) have inferred an equation for the binary recombination from further experiments (Eq. (28)), which was later adapted by Bates (1982) (Eq. (29)):

$$\alpha_2 = 6.8 \cdot 10^{-7} \cdot T^{-0.4} \text{ and} \tag{28}$$

$$\alpha_2 = 5 \cdot 10^{-8} \cdot \left(\frac{300}{T}\right)^{0.5}. \tag{29}$$

Furthermore, Smith and Adams (1982) presented a parameterisation valid for the altitude range of 10 to 60 km based on the laboratory measurements of binary recombination with the flowing afterglow/Langmuir probe (FALP) technique. The resulting parameterisation is simple because it only depends on the altitude and contains two terms that represent the ternary and binary recombination, respectively, as Eq. (30) shows:

$$\alpha = 1.63 \cdot 10^{-5} \cdot e^{-\frac{h}{7.38}} + 5.25 \cdot 10^{-8}, \tag{30}$$

where $h$ is the altitude in km. Johnsen et al. (1994) later disputed their results because they found that the ion-collecting probes, as used by Smith and Adams, are not suitable to obtain data on ion-ion recombination coefficients in flowing-afterglow studies. Bates (1982) criticised that the binary and ternary recombination rates had been erroneously considered additive in previous works, stating that both processes are not independent of each other. Instead of the binary recombination rate $\alpha_2$, he defined the enhancement due to the binary channel, $\Delta\alpha_2$, and calculated $\alpha_3$, $\Delta\alpha_2$, and the resulting total recombination coefficient, $\alpha$, in a Monte Carlo simulation for altitudes between 0 and 40 km. Interestingly, Smith, Church, Adams and Bates have never cited the works of Israël (1957) or Lenz (1932). It seems that the two latter authors have been overseen, probably because they published their works in German (however, Israël's textbook was translated into English in 1971). This is especially remarkable as Bates (1982) determined $\alpha$ to be $1.67 \cdot 10^{-6}$ $cm^3$ $s^{-1}$ at ground level which is in striking agreement with Israël (1957) ($1.6 \cdot 10^{-6}$ $cm^3$ $s^{-1}$) and Lenz (1932) (($1.7\pm0.1$) $\cdot 10^{-6}$ $cm^3$ $s^{-1}$). Instead, Bates referred to Sayers (1938) ($2.3 \cdot 10^{-6}$ $cm^3$ $s^{-1}$) and Nolan (1943) ($1.4 \cdot 10^{-6}$ $cm^3$ $s^{-1}$) whom he thought to be the first experimenters to quantitatively and accurately determine $\alpha$. By the beginning of the 1980s, science was in urgent need of correct and altitude-resolved values for the recombination coefficient. Arnold and Fabian (1980) presented a method to calculate the concentration of gaseous sulfuric acid from measured

concentration ratios of the ambient $HSO_4^-$ and $NO_3^-$ ions. The recombination coefficient, which describes the sink for ions, forms part of the formula (see Arnold and Qiu (1984) for a more detailed derivation). Until the early 1980s, this method was the only way to determine the concentration of trace gases, such as sulfuric or nitric acid, in the different layers of the atmosphere.

The need for an experimental investigation in the atmosphere was answered by Gringel et al. (1978), Rosen and Hofmann (1981), and Morita (1983). Gringel et al. (1978) conducted balloon-borne measurements of the air conductivity between 4 and 25 km over northern Germany in August and October 1976. From the measured air conductivity, $\sigma$, the calculated altitude-corrected ion mobility, $\mu$, and the mean of formerly measured ionisation rates, they determined the ion-ion recombination coefficients for different altitudes according to Eq. (31) and (32):

$$\alpha = \frac{q \cdot e^2 \cdot \mu^2}{\sigma^2}, \text{ where} \tag{31}$$

$$\mu = \mu_0 \cdot \frac{p_0}{p} \cdot \frac{T}{T_0}, \tag{32}$$

with $\mu_0 = 1.3 \cdot 10^{-4} \text{ m}^2 \text{ V}^{-1} \text{ s}^{-1}$. The altitude-resolved values for $q$ are the means of three independent measurements between the 1930s and the 1970s, although $q$ varies with the 11-year solar cycle which casts doubt on the validity of the values in the calculations.

Both Rosen and Hofmann (1981) and Morita (1983), on the other hand, measured the positive ion number concentration $n_+$ directly along with the ionisation rate $q$ in a concerted measurement campaign. Thus, Rosen and Hofmann's data, being available earlier than Morita's, were considered to be the most reliable ones at that time because they measured the relevant parameters simultaneously (Arijs, 1983). Applying Eq. (2), they calculated $\alpha$ for different altitude levels. In addition, for altitudes above 32 km, they used the alternative method given in Eq. (31) to calculate $\alpha$. The derived data points followed a profile suggested by a theory that accounted for both binary and ternary recombination. The data points derived with Eq. (31) fit the theoretical predictions better. However, below 9 km, the derived values for the ion-ion recombination were unexpectedly large. As the authors themselves wrote, in the troposphere, the losses of ions due to aerosol particle attachment have to be taken into account, otherwise the loss due to ion-ion recombination is overestimated when using Eq. (2); this is why only the values above 9 km are reliable. Nevertheless, these measurements have led to further adjustments of the parameterisations, such as the ones by Arijs et al. (1983) and Brasseur and Chatel (1983). Arijs et al. (1983) presented a formula that contains two terms, accounting for binary and ternary reactions, as shown in Eq. (33):

$$\alpha = 6 \cdot 10^{-8} \cdot \left(\frac{300}{T}\right)^{0.5} + 1.25 \cdot 10^{-25} \cdot [M] \cdot \left(\frac{300}{T}\right)^4, \tag{33}$$

where [M] is the number density of air molecules in $cm^{-3}$ (representing the pressure dependence), given by Eq. (34):

$$[M] = 7.243 \cdot 10^{18} \cdot \left(\frac{p}{T}\right). \tag{34}$$

Independently, Brasseur and Chatel (1983) proposed a very similar formula that only differs in the factor of the ternary recombination term (Eq. (35)):

$$\alpha = 6 \cdot 10^{-8} \cdot \left(\frac{300}{T}\right)^{0.5} + 6 \cdot 10^{-26} \cdot [M] \cdot \left(\frac{300}{T}\right)^4. \tag{35}$$

Due to the $T^{-1}$ dependence of [M], the ternary recombination coefficient ultimately shows a $T^{-5}$ dependence in Arijs et al. (1983) and Brasseur and Chatel (1983), which is even stronger than previously discussed. The temperature dependence of $T^{-0.5}$ in the binary term is in agreement with Hickman (1979) (but not with the more recent parameterisation of Miller et al. (2012) that describes a $T^{-0.9}$ dependence). The pressure dependence of $\alpha$ is $p^1$ in the ternary terms in Eq. (33) and (35) through the linear $p$ dependence of [M] (see Eq. (34)). A proportional pressure dependence is also observed in some Thomsonian theories

(see Sect. 2).

    Parallel to Rosen and Hofmann, Morita (1983) conducted atmospheric field measurements and also found reasonable results for altitudes above 6 km; however, for the above-mentioned reason, strong disagreement of the observed results from the theoretically expected ones below 6 km was found. Bates (1985) synthesised his earlier model results (see Sect. 6) and the measurements by Morita to define another parameterisation that is merely dependent on the altitude, as shown in Eq. (36):

$\alpha = 5.33 \cdot 10^{-6} \cdot e^{-0.111 \cdot h}$,                                                                  (36)

    which is valid for the range of 10 to 25 km. Below 10 km, $\alpha$ is expected to be constant at $1.7 \cdot 10^{-6}$ cm$^3$ s$^{-1}$.

    More recently, Franchin et al. (2015) reported experimental data for the recombination coefficient for atmospheric conditions. They have examined the dependency of $\alpha$ on the temperature, relative humidity $RH$, and the number concentrations of sulfur dioxide, [SO$_2$], and ozone, [O$_3$], in a series of chamber experiments. Their experimental setup did not allow for pressures below

1013 hPa, therefore, it is not directly possible to derive new insights with regard to processes in the upper troposphere or stratosphere. However, they did show that $\alpha$ is dependent on $RH$; with increasing relative humidity, the recombination coefficient decreases. At 70 % $RH$, $\alpha$ is $2.0 \cdot 10^{-6}$ cm$^3$ s$^{-1}$ which is within the known range of uncertainty; however, for 7 % $RH$, it is as high as $8.1 \cdot 10^{-6}$ cm$^3$ s$^{-1}$ (both at 293.15 K). They convincingly explain this by an increase in ion sizes with increasing $RH$. With a higher humidity, more water ligands are added to the ion cluster and, thus, its size and mass increases

while its mobility decreases. This effect could not be reproduced quantitatively by any theory (Franchin et al., 2015). Their data also revealed the temperature dependence of $\alpha$: at 293.15 K, the value was $(2.3 \pm 0.7) \cdot 10^{-6}$ cm$^3$ s$^{-1}$ and, at 218.15 K, it was $(9.7 \pm 1.2) \cdot 10^{-6}$ cm$^3$ s$^{-1}$ (both at 40 % $RH$). Unfortunately, the standard deviations of the data points are large, thus, any temperature dependence derived from the four data tuples is inaccurate in itself. Nevertheless, we derived a temperature dependence of $T^{-3}$ from their data. This is in a similar order of magnitude to the findings of Smith and Church (1977) ($T^{-2.5}$ to

$T^{-3}$) and is still comparable to Arijs et al. (1983) and Brasseur and Chatel (1983) (both teams: $T^{-5}$) for the ternary recombination, considering that ternary recombination is the predominant process at atmospheric pressure. Besides this, the recombination coefficient was found to be independent of [O$_3$] and [SO$_2$].

    After 1985, no further improvements of the parameterisation for direct application in the atmosphere have been made. One reason could be that the need for this value has become less urgent, since, from that year, trace gases could be measured directly

in their neutral forms (Arnold and Hauck, 1985). In addition, most of the parameterisations were in good accordance for the altitude range of 10 to 40 km (Arijs, 1983) so that no further improvement appeared to be necessary. As yet, for the troposphere, experimental validation of the parameterisations remains open until this day. The best estimate available is the assumption that $\alpha$ remains constant between 0 and 10 km due to the cancellation of the opposing temperature and pressure effects. However,

theories concerning the ion-aerosol attachment have been further developed. The most important theories and their
applicability to the ion-ion recombination will be discussed in the next section.

## 5 Application of ion-aerosol theories

Apart from ion-ion recombination, the analogous process of ion-aerosol attachment was also further studied. For instance, Natanson (1960a) (English translation of the original in Russian: Natanson (1960b)) expanded his approach to the attachment of ions to particles. In general, the ion-ion recombination can be considered as a special case of the ion-aerosol attachment, i.e.
when the radius of the aerosol particle is reduced to ionic sizes. While in many ion-ion recombination theories, the concept of the three-body collision radius, or trapping radius, $d$, can be found, many ion-aerosol theories additionally use the concept of the limiting sphere, $\delta$. The limiting sphere and its radius are defined slightly differently depending on the theory. With Fuchs (1963), it is defined as a concentric sphere around the particle with the radius $\delta_F = r_p + \lambda'$, where $r_p$ is the particle radius and $\lambda'$ is "the mean distance from the surface of the particle at which the ions collide for the last time with gas molecules before
striking this surface" (Fuchs, 1963). Notably, $\lambda'$ is not equal to the mean free path of one ion, $\lambda_{ion}$, or the ion-ion mean free path, $\lambda$. With Hoppel and Frick (1986), it is defined as the sum of the ion-aerosol three-body trapping sphere and the ion-ion mean free path (see Eq. (44)). Transferred to the ion-ion recombination, the limiting sphere can be defined as the sum of the ion-ion three-body trapping distance and one mean free path (see Eq. (45)), as depicted in Fig. 1.

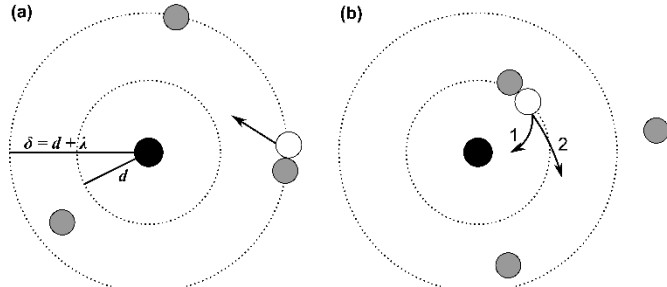

**Figure 1: Schematic of the limiting sphere, based on Hoppel and Frick (1986), López-Yglesias and Flagan (2013) and Tamadate et al. (2020b). The black circle in the centre represents an ion and the white circle represents an ion of opposite sign, while the grey circles represent neutral gas molecules. The inner dotted circle with the radius $d$ is the ion-ion trapping sphere, while the outer dotted circle with the radius $\delta = d + \lambda$ describes the limiting sphere. The spheres are defined differently in various theories; this schematic depicts the theory of Hoppel and Frick (1986). a) The approaching white ion experiences its last collision with a neutral**
**gas molecule outside the ion-ion trapping sphere; here, one mean free path away from the surface of the trapping sphere. b) The white ion collides with another neutral gas molecule, this time leading to it entering the trapping sphere, i.e. being "trapped". In case 1, it collides and subsequently recombines with the black ion of opposite sign. In case 2, no ion-ion collision and recombination occur and the white ion leaves the trapping sphere.**

In Fig. 1 (a), an ion (white circle) approaches the ion in the centre (black circle) which has the opposite charge. The approaching
ion experiences its last collision with a neutral gas molecule approximately one mean free path away from the trapping sphere, (i.e. on the surface of the limiting sphere according to Hoppel and Frick's definition). When entering the limiting sphere, the

white ion collides with another neutral gas molecule on the surface of the trapping sphere. This process is also called three-body trapping, because the gas molecule, which is the third body, "traps" the white ion inside the trapping sphere of the centre ion. However, not all of these collisions lead to the recombination of the two ions. The probability for an ion to collide with a

neutral gas molecule is accounted for in many theories. Two possible outcomes of the third-body collision are shown as cases 1 (collision) and 2 (non-collision) in Fig. 1 (b).

Tamadate et al. (2020b) provided a comprehensive review of those theories and their application to the ion-ion recombination in the introductory part of their work. For the sake of completeness and to identify the potential of these approaches for their application to the lower atmosphere, the most important concepts and formulae are given in the following paragraphs.

In Fuchs's theory, outside the limiting sphere, continuum equations are used, whereas inside the sphere, kinetic theory is applied because steady-state, rather than equilibrium conditions, are valid (Fuchs, 1963; Hoppel and Frick, 1986). Coulomb as well as image forces are taken into account in this theory, but no third-body processes. The radius $\delta_F$ of the limiting sphere is given in Eq. (37) (Fuchs, 1963), based on considerations by Wright (1960):

$$\delta_F = \frac{r_{coll}^3}{\lambda^2} \cdot \left[ \frac{1}{5} \cdot \left(1 + \frac{\lambda}{r_{coll}}\right)^5 - \frac{1}{3} \cdot \left(1 + \frac{\lambda^2}{r_{coll}^2}\right) \cdot \left(1 + \frac{\lambda}{r_{coll}}\right)^3 + \frac{2}{15} \cdot \left(1 + \frac{\lambda^2}{r_{coll}^2}\right)^{2.5} \right],$$ (37)

where $r_{coll}$ is the collision radius, which is the sum of both ion radii (originally, the particle radius $r_p$).

Although Fuchs (1963) did not provide a formula for $\alpha$ himself, Tamadate et al. (2020b) have used Fuchs's approach to derive the ion-ion recombination coefficient, shown in Eq. (38) to (40):

$$\alpha = \frac{(D_+ + D_-) \cdot e^2}{\varepsilon_0 \cdot k_B T \cdot \left[1 - \exp\left(\frac{-e^2}{4\pi \cdot \varepsilon_0 \cdot k_B T \cdot \delta_F}\right)\right]} \cdot \left(1 + \frac{(D_+ + D_-) \cdot e^2}{\alpha_\delta \cdot \varepsilon_0 \cdot k_B T \cdot \left[\exp\left(\frac{e^2}{4\pi \cdot \varepsilon_0 \cdot k_B T \cdot \delta_F}\right) - 1\right]}\right)^{-1}, \text{ where}$$ (38)

$$\alpha_\delta(r_{coll}) = \pi \cdot r_{coll}^2 \cdot \left(\frac{8k_B T}{\pi \cdot m_{red}}\right)^{0.5} \cdot \gamma(r_{coll}), \text{ and}$$ (39)

$$\gamma(r_{coll}) = 1 + \frac{e^2}{4\pi \cdot \varepsilon_0 \cdot k_B T} \cdot \left(\frac{1}{r_{coll}} - \frac{1}{\delta_F}\right),$$ (40)

where $\alpha_\delta$ is the ion-ion collision rate coefficient at the limiting sphere surface.

However, there are two main problems with the theory of Fuchs (1963). Firstly, it ignores the three-body trapping, i.e. collisions of the two ions (or the ion and the particle) with neutral gas molecules (Hoppel and Frick, 1986; Tamadate et al., 2020b). Hoppel and Frick (1986) showed that Fuchs's theory would lead to an ever-growing underestimation of the attachment

coefficient for decreasing particle radii (e.g. $4.4 \cdot 10^{-7}$ cm$^3$ s$^{-1}$ for a 1 nm radius) due to the negligence of the three-body trapping. Three-body trapping becomes increasingly more relevant for aerosol particles approaching ionic sizes (or when two ions recombine) and when the pressure is relatively high, as is the case in the lower atmosphere. Secondly, an ion entering the limiting sphere of another ion with the opposite sign does not follow the thermal equilibrium distribution; instead, the equilibrium ion drift velocity, $v = \mu E$, in the direction of the ion in the center of the limiting sphere needs to be considered

(Gopalakrishnan and Hogan, 2012; Tamadate et al., 2020b).

Similarly, Tamadate et al. (2020b) derived $\alpha$ using the theory of Filippov (1993) who had examined the ionic charging of small aerosol particles with respect to the Knudsen number, which was, again, based on the considerations of Fuchs (1963). Tamadate

et al. (2020b) presented Eq. (41) that defines the ion-ion recombination coefficient and which already includes the collision probability for ions entering the limiting sphere, $\varepsilon_\delta$, and is independent of $\alpha_\delta$:

$$\alpha = \frac{4\pi \cdot (D_+ + D_-) \cdot \delta \cdot \Psi_\delta}{1 - \exp(-\Psi_\delta)} \cdot \left[ 1 + \left( \frac{\pi}{2} \right)^{0.5} \cdot \frac{2 - \varepsilon_\delta}{\varepsilon_\delta} \cdot Kn_\delta \cdot \frac{\Psi_\delta}{\exp(\Psi_\delta) - 1} \right]^{-1}, \text{ with} \tag{41}$$

$$\Psi_\delta = \frac{e^2}{4\pi \cdot \varepsilon_0 \cdot k_B T \cdot \delta}, \text{ and} \tag{42}$$

$$Kn_\delta = \left( \frac{m_{\text{red}}}{k_B T} \right)^{0.5} \cdot \frac{D_+ + D_-}{\delta}, \tag{43}$$

where $\Psi_\delta$ is the ratio of Coulomb and thermal energy at the limiting sphere surface and $Kn_\delta$ is the Knudsen number for the limiting sphere. To obtain $\varepsilon_\delta$, Tamadate et al. (2020b) performed molecular dynamics (MD) simulations that will be described in the next section.

Subsequent to Fuchs, Hoppel and Frick (1986) developed a theory for ion-aerosol attachment based on Natanson (1960), Keefe et al. (1968), and Hoppel (1977) that uses the limiting sphere approach and includes image forces and three-body trapping. Hoppel and Frick (1986) defined limiting sphere radii for both effects as well as for their combination, which indicate the maximum distance for which the ions would recombine with a particle given the respective effect(s). Since the collisions with a third body (i.e. a gas molecule) only occur with a certain probability within the limiting sphere, this probability needs to be taken into account in a similar way as in the Thomson theory. Hoppel and Frick (1986) show that image forces are not relevant for the case of ion-ion recombination or for ion-particle attachment when the particle diameter is small ($< 40$ nm). For this condition, they define the limiting sphere radius, $\delta_{\text{HF}}$, as the sum of the ion-aerosol three-body trapping distance, $d_{\text{ia}}$, and the ion-ion mean free path according to Eq. (44):

$$\delta_{\text{HF}} = d_{\text{ia}} + \lambda, \tag{44}$$

The value for the attachment coefficient of a singly charged particle smaller than 2 nm in radius and an ion of opposite sign reported by Hoppel and Frick (1986) approaches $1.3 \cdot 10^{-6}$ cm$^3$ s$^{-1}$ and, thus, approaches the ion-ion recombination coefficient itself. Thus, by applying Eq. (44) to ion-ion recombination, one can derive Eq. (45) where the ion-aerosol trapping sphere $d_{\text{ia}}$ is replaced by the ion-ion trapping sphere $d_{\text{HF}}$ (see also Tamadate et al. (2020b)):

$$\delta_{\text{HF}} = d_{\text{HF}} + \lambda. \tag{45}$$

In the following we want to outline briefly how the Hoppel and Frick method is used to determine the ion-aerosol attachment coefficients. Their method does not provide any means to calculate the ion-aerosol trapping distance accurately from theory. Therefore, they adopt the theory by Natanson (1959) to derive the ion-ion trapping distance from a measured ion-ion recombination coefficient and certain ion properties (they take the value of $\alpha = 1.4 \cdot 10^{-6}$ cm$^3$ s$^{-1}$ from Nolan (1943)). The value of the ion-ion trapping sphere distance can then be used to calculate the ion-aerosol trapping sphere distance. Since the method by Hoppel and Frick (1986) was explicitly developed to determine ion-aerosol attachment coefficients, it is not directly suitable to determine ion-ion recombination coefficients. One important application for the Hoppel and Frick theory is the calculation of aerosol equilibrium charge distributions as function of the particle diameter. The knowledge of the charged fractions (as a function of diameter and the number of elementary charges) are important for aerosol size distribution

measurements with differential mobility analyzers after the aerosol is "neutralised" by passing a strong ion source with high concentrations of bipolar ions (e.g., Wang and Flagan, 1990). The fact that the method by Hoppel and Frick (1986) does not include any means of calculating the ion-aerosol or the ion-ion trapping distance directly was also discussed by Tamadate et al. (2020b). They highlight that the effect of changing pressure and temperature on the trapping distance is not taken into account. Nevertheless, López-Yglesias and Flagan (2013) have improved some approximations made by Hoppel and Frick (1986) and developed a model to calculate the ion-aerosol attachment for aerosol particles of different sizes and charges. The amendment of using Maxwellian speed distributions for the ion and the colliding gas molecule instead of fixed average speeds led to the most significant of the deviations from Hoppel and Frick's model.

Tamadate et al. (2020b) provided a set of formulae in order to calculate the ion-ion recombination rate after Hoppel and Frick, also using Eq. (38). However, $\alpha_\delta$, the ion-ion collision rate coefficient at distance $\delta$, is defined differently for Hoppel and Frick, and is given in Eq. (46). Here, the ion-ion collision rate $\alpha_\delta$ is not directly dependent on the collision radius $r_{coll}$ (cf. Eq. (39)), but on the ion-ion trapping distance $d_{HF}$. Furthermore, the additional probability factor for ions entering the trapping sphere, $\varepsilon_d$, is introduced to the ion-ion collision rate:

$$\alpha_\delta(d_{HF}) = \pi \cdot d_{HF}^2 \cdot \left(\frac{8k_BT}{\pi \cdot m_{red}}\right)^{0.5} \cdot \gamma(d_{HF}) \cdot \varepsilon_d, \text{ with} \tag{46}$$

$$\gamma(d_{HF}) = 1 + \frac{e^2}{4\pi \cdot \varepsilon_0 \cdot k_BT} \cdot \left(\frac{1}{d_{HF}} - \frac{1}{\delta_F}\right), \tag{47}$$

$$\varepsilon_d = 1 - \frac{\lambda^2}{2 \cdot d_{HF}^2} \cdot \left(1 - \exp\left(\frac{-2 \cdot d_{HF} \cdot \cos\theta}{\lambda}\right) \cdot \left(1 + \frac{2 \cdot d_{HF}}{\lambda} \cdot \cos\theta\right)\right), \tag{48}$$

$$\theta = \sin^{-1}\left(\frac{b}{d_{HF}}\right), \text{ and} \tag{49}$$

$$b = r_{ion} \cdot \sqrt{1 + \frac{e^2}{32 \cdot k_BT \cdot \varepsilon_0} \cdot \left(\frac{1}{r_{coll}} - \frac{1}{d_{HF}}\right)}, \tag{50}$$

where $\theta$ is the critical angle to enter the trapping sphere and $b$ is the critical collision parameter (Tamadate et al., 2020b). Again, to obtain a value for $\alpha$, one needs to know the given trapping sphere radius, $d_{HF}$. However, to determine $d_{HF}$, Hoppel and Frick used a known ion-ion recombination coefficient. This circular logic arises because we divert their theory that is meant to determine ion-aerosol attachment processes to ion-ion recombination processes. Nevertheless, it can be tested to calculate $\alpha$ for different altitudes of the atmosphere by keeping a constant value for $d_{HF}$ while varying $T$ and $p$.

## 6 Numerical simulations

Tamadate et al. (2020b) highlight that there is no single calculation approach that yields accurate ion-ion recombination rates for a wide range of pressures, temperatures, gas compositions, and ion chemical compositions. This deficiency motivated their development of a so-called hybrid continuum-molecular dynamics (MD) approach. This method couples the limiting sphere method, when the two ions are sufficiently far apart from each other and their motion is controlled by diffusion, with MD simulations that model the ion motions within a critical distance $\delta$. The calculations are applied to a system where $NH_4^+$ and

$NO_2^-$ ions recombine in helium at 300 K under varying pressure. Collisions between the ions and neutral gas molecules are taken into account. Excellent agreement is found when the model results are compared with the laboratory measurements at two different pressures (Lee and Johnsen, 1989). The equation for calculating the ion-ion recombination coefficient (Eq. (41)) is derived from Filippov (1993). For their test case, Tamadate et al. (2020b) show that the limiting sphere distance suggested by Fuchs (1963) ($\delta_F$, see Eq. (37)) can be used as the initial distance between the two ions when the MD simulations commence. For larger distances, the calculated recombination rates do not change which demonstrates that the proposed method is independent on the choice of the limiting sphere radius, as long as it is sufficiently large. The quantity that is determined by the simulations is the probability, $\varepsilon_\delta$, that a successful collision occurs when the initial speeds of the ions in 3D are drawn from probability density functions based on Boltzmann distributions. A collision is defined as being successful if the distance between the two ions gets smaller than a threshold value, whereas it is not successful if the distance eventually exceeds $\delta_F$. Especially for low pressures (and correspondingly large $\delta_F$), $\varepsilon_\delta$ can become very small and, thus, requires many simulations for achieving results with small statistical errors. The MD simulations require, in addition, Lennard-Jones parameters and partial charges on atoms as input variables. In a separate publication, Tamadate et al. (2020a) apply their continuum-MD approach to a system where positively charged polyethylene glycol ions (1 to 7 charges, mass of 4600 Da) recombine with $NO_2^-$ ions in nitrogen. The comparison between the experimentally determined recombination rates and the calculated ones indicates that they agree within a factor of two. These results show that the hybrid continuum-MD approach is well suited to yield accurate ion-ion recombination rates for a wide range of applications and conditions, including studies of the different layers of the atmosphere.

Numerical simulations using Monte Carlo (MC) methods have been another powerful tool to gain insights into ion-ion recombination or ion-particle attachment rates and their dependencies on parameters such as gas pressure and temperature. In contrast to the MD simulations, the ion-ion and the ion-neutral interactions are generally much simplified, for example, the collisions with neutral gas molecules are treated by the use of random numbers for the collision frequencies, energies, and angles. In the majority of cases, the collisions are treated as elastic, while spherical geometry is assumed for the collision partners. The first MC calculations to include three-body trapping were conducted by Feibelman (1965) who found good agreement with a measured recombination rate. Later MC simulations studied the recombination in oxygen for varying pressure from zero pressure, i.e., for the binary condition, up to approximately 1000 hPa where ternary recombination is clearly dominant (Bates and Mendaš, 1978). The results showed that the ion-ion recombination rates peak between 1000 and 2000 hPa as expected from theory. Besides the pressure dependence of the recombination rate, Bardsley and Wadehra (1980) also studied the temperature dependence using MC simulations. The results indicate a stronger than $T^{-3}$ dependence for low pressures and a strongly reduced temperature dependence for pressures above approximately 5000 hPa. Bates (1982) reported ion-ion recombination rates calculated with an MC model for the atmospheric conditions of between 0 and 40 km altitude, which agree well when compared to the values from the balloon measurements (see Sect. 7.1). Filippov (1993) developed an MC model

for the charging of aerosol particles. The numerical results show fairly good agreement with the measured values in the range of between 5 and 80 nm when using either air or helium as the neutral gas.

## 7 Comparison of the parameterisations and theories

In Table 2, all theories, parameterisations, field and laboratory data sets, and model results discussed in the previous sections
are listed for a better overview. In the following sections, they will be addressed by the abbreviations listed in Table 2. In order to determine the most suitable single formula to determine the ion-ion recombination coefficient for different altitudes, we compare the above mentioned parameterisations and theories to the field, laboratory and model data.

### 7.1 Comparison to field and model data

In a first step, the parameterisations and theories were compared to the field and model data. The temperature, pressure and air
density data of the US Standard Atmosphere were used here (National Oceanic and Atmospheric Administration et al., 1976). Furthermore, we used the parameters $m_{ion} = m_+ = m_- = 90$ Da, $\mu_0 = 1.35 \cdot 10^{-4}$ m$^2$ V$^{-1}$ s$^{-1}$ (López-Yglesias and Flagan, 2013), $d_{HF} = 18$ nm for HF86 (the proposed value of Hoppel and Frick (1986) for an ion of 90 Da, assuming an ion-ion recombination coefficient of $1.7 \cdot 10^{-6}$ cm$^3$ s$^{-1}$), and $EA = 3.94$ eV (Weaver et al., 1991). $r_{coll}$ was calculated according to Eq. (A1). The results are plotted in Fig. 2. Here, the y-axes represent the altitude $h$ and the x-axes represent the ion-ion recombination coefficient $\alpha$.
In Fig. 2 (a) to (d), the field measurements Gr78, RH81, and Mo83 are shown for a better comparability. Note that the data are inaccurate below 10 km. For RH81, there are two data sets for altitudes above 32 km: one is calculated based on Eq. (2), the other one is based on Eq. (31). One should bear in mind that these data sets, which were determined with similar methods, may also suffer from systematic errors such as losses inside the instrument that were not accounted for, however, these remain the most reliable data from field measurements available to this day. The challenge for the theories and parameterisations is to
accurately determine the ion-ion recombination coefficient for the different regimes: the ternary recombination regime between 0 and approximately 25 km, the transition regime between 25 and approximately 40 km, and the binary recombination regime above 40 km. Note that the binary theories (i.e. Fu63, Hi79, and Mi12) are, therefore, only plotted above 40 km altitude. In Fig. 2 (a), the Thomsonian theories of Ga38, Lo60, Is57, and Na59 are depicted. In Fig. 2 (b), the semi-empirical adjustments to the Thomson theory Ar83 and BC83 as well as the Monte Carlo simulation Ba82 are shown and in Fig. 2 (c), the binary
complex potential models Hi79 and Mi12 as well as the solely altitude-dependent parameterisations SA82 and Ba85 are plotted. Figure 2 (d) shows Fu63 and HF86, the ion-aerosol attachment theories that are applied to the ion-ion recombination. Figure 2 (e) provides an overview of the most relevant theories and datasets for an altitude range of up to 12 km with a linear x-scale, whereas the other subplots use a logarithmic x-scale.

 **Table 2: List of all theories, parameterisations, data sets, and models used for comparison.**

| Study | Abbr. | Formula (for parameterisations) / Range (for data sets) | |
|---|---|---|---|
| *Theories and parameterisations* | | | |
| Gardner, 1938 | Ga38 | $\alpha = 1.9 \cdot 10^{-5} \cdot \left(\frac{273}{T}\right)^{1.5} \cdot \left(\frac{1}{m_{ion}}\right)^{0.5} \cdot \varepsilon_T(T,p)$ | Eq. (10) to (13) |
| Loeb, 1955/1960 | Lo60 | $\alpha = 1.73 \cdot 10^{-5} \cdot \left(\frac{273}{T}\right)^{1.5} \cdot \left(\frac{1}{m_{ion}}\right)^{0.5} \cdot \varepsilon_T(T,p)$ | Eq. (14) and (11) to (13) |
| Israël, 1957/1971 | Is57 | $\alpha = 1.95 \cdot 10^{-6} \cdot \left(\frac{273}{T}\right)^{1.5} \cdot \varepsilon_T(T,p)$ | Eq. (15) and (11) to (13) |
| Natanson, 1959 | Na59 | $\alpha = \dfrac{\pi \cdot d_N^2 \cdot v_{rel} \cdot \varepsilon_N \cdot \left[1 + \frac{e^2 \cdot \lambda_{ion}}{4\pi \cdot \varepsilon_0 \cdot d_N \cdot (d_N + \lambda_{ion}) \cdot k_B T}\right] \cdot \exp\left(\frac{e^2}{4\pi \cdot \varepsilon_0 \cdot (d_N + \lambda_{ion}) \cdot k_B T}\right)}{1 + \frac{\pi \cdot \varepsilon_0 \cdot d_N^2 \cdot v_{rel} \cdot \varepsilon_N \cdot k_B T}{e^2 \cdot D} \cdot \left[1 + \frac{e^2 \cdot \lambda_{ion}}{4\pi \cdot \varepsilon_0 \cdot d_N \cdot (d_N + \lambda_{ion}) \cdot k_B T}\right] \cdot \left[\exp\left(\frac{e^2}{4\pi \cdot \varepsilon_0 \cdot (d_N + \lambda_{ion}) \cdot k_B T}\right) - 1\right]}$ | Eq. (16) to (21) |
| Hickman, 1979 | Hi79 | $\alpha_2 = 5.35 \cdot 10^{-7} \cdot \left(\frac{T}{300}\right)^{-0.5} \cdot m_{red}^{-0.5} \cdot EA^{-0.4}$ | Eq. (25) |
| Miller et al., 2012 | Mi12 | $\alpha_2 = 2.8 \cdot 10^{-7} \cdot \left(\frac{T}{300}\right)^{-0.9} \cdot m_{red}^{-0.5} \cdot EA^{-0.13}$ | for polyatomic anions, Eq. (27) |
| Smith and Adams, 1982 | SA82 | $\alpha = 1.63 \cdot 10^{-5} \cdot e^{-\frac{h}{7.38}} + 5.25 \cdot 10^{-8}$ | valid from 10 to 60 km, Eq. (30) |
| Arijs et al., 1983 | Ar83 | $\alpha = 6 \cdot 10^{-8} \cdot \left(\frac{300}{T}\right)^{0.5} + 1.25 \cdot 10^{-25} \cdot [M] \cdot \left(\frac{300}{T}\right)^{4}$ | Eq. (33) and (34) |
| Brasseur and Chatel, 1983 | BC83 | $\alpha = 6 \cdot 10^{-8} \cdot \left(\frac{300}{T}\right)^{0.5} + 6 \cdot 10^{-26} \cdot [M] \cdot \left(\frac{300}{T}\right)^{4}$ | Eq. (35) and (34) |
| Bates, 1985 | Ba85 | $\alpha = 5.33 \cdot 10^{-6} \cdot e^{-0.111 \cdot h}$ | valid from 10 to 25 km, Eq. (36) |
| Fuchs, 1963 | Fu63 | $\alpha = \dfrac{\dfrac{(D_+ + D_-) \cdot e^2}{\varepsilon_0 \cdot k_B T \cdot \left[1 - \exp\left(\frac{-e^2}{4\pi \cdot \varepsilon_0 \cdot k_B T \cdot \delta_F}\right)\right]}}{1 + \dfrac{(D_+ + D_-) \cdot e^2}{\alpha_\delta \cdot \varepsilon_0 \cdot k_B T \cdot \left[\exp\left(\frac{e^2}{4\pi \cdot \varepsilon_0 \cdot k_B T \cdot \delta_F}\right) - 1\right]}}$ | Eq. (38) to (40) and (37) |
| Hoppel and Frick, 1986 | HF86 | see Fu63 | Eq. (38), (37) and (46) to (50) |
| *Field data* | | | |
| Gringel et al., 1978 | Gr78 | $\alpha$ derived from measurements of $q$ and $\sigma$ at 4 to 30 km | |
| Rosen and Hofmann, 1981 | RH81 | $\alpha$ derived from measurements of $q$ and $n_+$ at 2 to 36 km and of $q$ and $\sigma$ at 32 to 45 km | |
| Morita, 1983 | Mo83 | $\alpha$ derived from measurements of $q$ and $n_+$ at 3 to 35 km | |
| *Laboratory data* | | | |
| Franchin et al., 2015 | Fr15 | $\alpha(T)$ for $RH$ = const. and $\alpha(RH)$ for $T$ = const.; $\alpha \sim T^{-3}$ (approximately) | |
| *Model data* | | | |
| Bates, 1982 | Ba82 | Monte Carlo simulation of $\alpha_3$, $\Delta\alpha_2$ and $\alpha$ for 0 to 40 km in 5 km steps | |

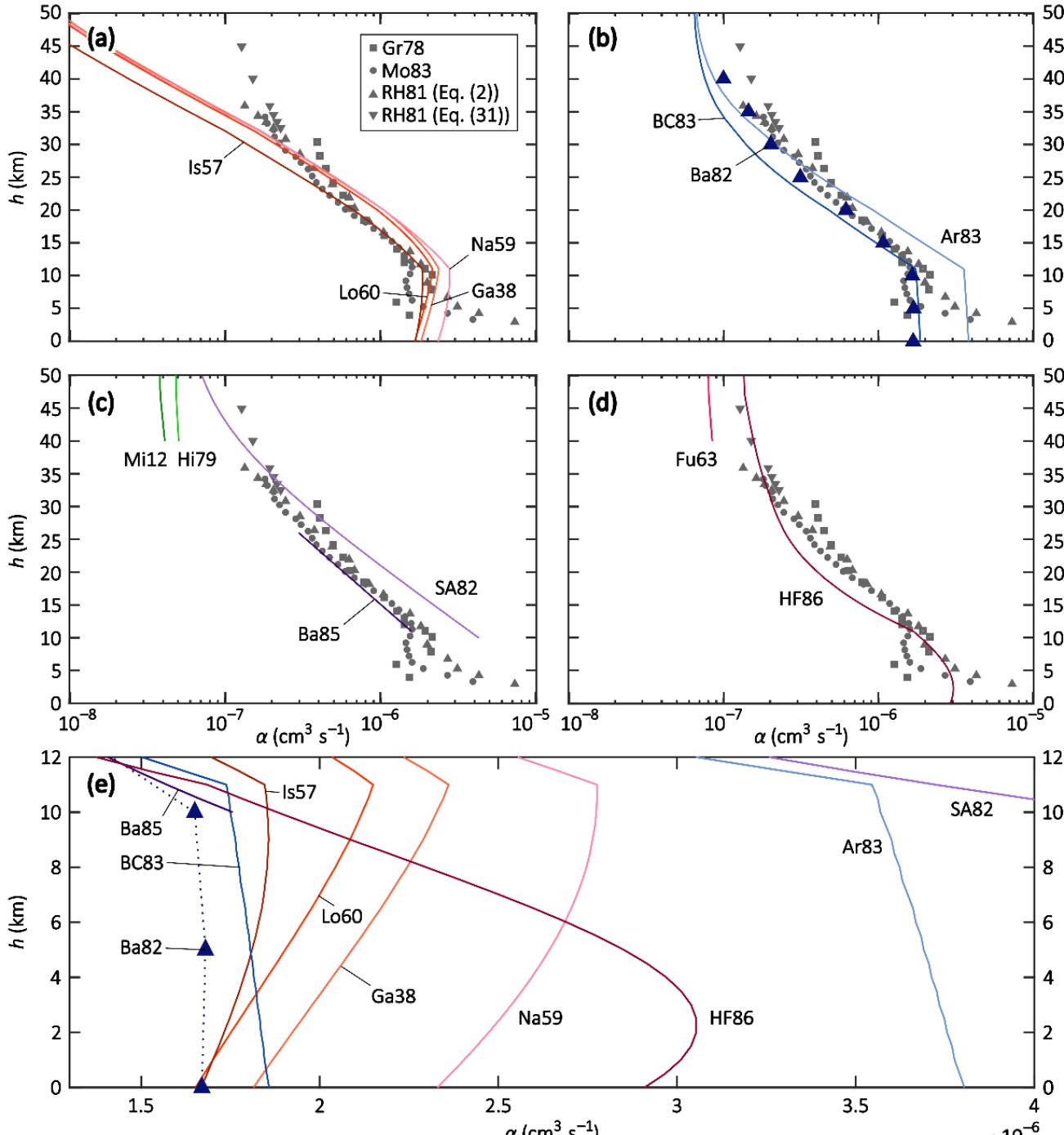

**Figure 2:** Altitude profiles of theories and parameterisations (solid lines), field data (grey symbols), and model simulations (purple triangles) of the recombination coefficient $\alpha$ for conditions of the US Standard Atmosphere. a) to d) Altitude profiles up to 50 km with a logarithmic x-axis; the field data are depicted in all panels for a better comparability. e) Altitude profiles up to 12 km with a linear x-axis. The meanings of the labels are listed in Table 2. See text for more details.


The Thomsonian theories (Ga38, Lo60, Is57, and Na59; see Fig. 2 (a) and (e)) all show a similar behaviour; from ground level up to 11 km (tropopause), the $\alpha$ value slightly increases and above 11 km, it decreases more strongly. Above 16 km, Ga38, Na59, and Lo60 yield almost the same values, whereas Is57 yields lower values throughout the stratosphere. Close to the ground, Ga38, Lo60, and Is57 predict $\alpha$ values identical or close to $1.7 \cdot 10^{-6}$ cm$^3$ s$^{-1}$, whereas it is slightly elevated for Na59.

Comparing these theories to the field data, Is57 shows a striking agreement for the altitude range of 11 to 22 km. However, none of the theories predict the slower decrease and asymptotic approach to a constant value due to the binary recombination predominating over the ternary process. Apparently, the binary process is not sufficiently taken into account in these theories, thus, their validity is limited to the altitude range of 0 to 22 km. However, within this range, the parameterisation of Is57 yields the most promising results.

The Monte Carlo simulation Ba82 (see Fig. 2 (b) and (e)) reproduces the ground level value of $1.7 \cdot 10^{-6}$ cm$^3$ s$^{-1}$ and yields almost constant values for 0, 5, and 10 km altitudes, while decreasing above 10 km, reproducing the field data with remarkable agreement. The semi-empirical parameterisations Ar83 and BC83 (see Fig. 2 (b) and (e)) contain both a binary and a ternary recombination term and were developed to reproduce the dataset of RH81; BC83 does so between 11 and approximately 20 km, while Ar83 reproduces the data set between approximately 20 and 35 km. In addition, BC83 predicts a ground level

value of $1.9 \cdot 10^{-6}$ cm$^3$ s$^{-1}$ for $\alpha$, which is much closer to the expected value than that of $3.8 \cdot 10^{-6}$ cm$^3$ s$^{-1}$ by Ar83. Close to 50 km, both parameterisations approach a similar value because their binary term is identical and becomes increasingly more dominant at higher altitudes. The strong deviations especially in the troposphere show that small changes in the prefactor of the ternary term can have substantial effects on the resulting recombination coefficient. For altitudes above 25 km, Ar83 can be chosen to parameterise the ion-ion recombination coefficient because it reproduces the field data sufficiently well. It is

worthwhile noting that BC83 and Ar83 have a strong temperature dependence of $T^{-5}$ while the Thomsonian theories have a much weaker temperature dependence of $T^{-1.5}$, although they do still yield similar results in the troposphere.

The solely altitude-dependent empirical parameterisations SA82 and Ba85 (see Fig. 2 (c) and (e)) are valid from 10 to 60 km and from 10 to 25 km, respectively. Ba85, indeed, reproduces the field data of RH81 well within this range. Sa82 overestimates the recombination coefficient below an altitude of 30 km but fits the field data reasonably well between 32 and 45 km; this is

because it features the slower decrease of $\alpha$ for increasing altitudes above 30 km where the binary recombination process predominates. Given their solely empirical nature and their limited validity ranges, these two parameterisations appear to be useful in applications where there is only information available on the altitude and knowledge on the temperature or pressure is lacking.

The two binary theories Hi79 and Mi12 (see Fig. 2 (c), only shown above 40 km altitude) yield lower recombination

coefficients than the field data, while still being in the same order of magnitude as SA82, Ar83, and BC83. However, as the data coverage is scarce above 35 km altitude, it is difficult to compare these theories to the field data and draw appropriate conclusions.

The two ion-aerosol attachment theories Fu63 and HF86 (see Fig. 2 (d) and (e)) when applied to the ion-ion recombination process do not reproduce the field measurements. While Fu63 only accounts for the binary recombination process and is,

therefore, only shown in the binary regime, i.e. above 40 km altitude, it yields reasonable results with values for $\alpha$ that are only slightly lower compared to the field data. HF86, on the other hand, yields results that are in the correct order of magnitude within the troposphere, however, the $\alpha$ value of $2.9 \cdot 10^{-6}\,\mathrm{cm^3\,s^{-1}}$ at ground level is higher than expected. Within the stratosphere, one can observe the increasing dominance of the binary recombination process in HF86 with increasing altitude since the curve approaches a constant value, as seen in other theories discussed above. The recombination coefficient is in

excellent agreement with the field data above 30 km altitude, however, it is underestimated between 15 and 25 km. One possible source for these inaccuracies could be the assumption of a constant value of 18 nm for the ion-ion trapping sphere radius $d_{HF}$; in other theories, this value is dependent on $T$ and/or $p$. An altitude-dependent trapping distance may improve the performance of HF86 (see Sect. 8).

In summarising the inter-comparison, it becomes obvious that there does not exist a single theory that can accurately reproduce

the known ground level value for the recombination coefficient $\alpha$ as well as the field data between 10 and 45 km altitude. Within the troposphere, it is reasonable to assume that the recombination coefficient stays almost constant due to the counterbalancing temperature and pressure effects. In the stratosphere, on the other hand, it decreases, although the decline lessens in the upper stratosphere, approaching an almost constant value at the top of the stratosphere. For altitudes between 0 and 22 km, the Thomsonian parameterisation Is57 reproduces the ground level data and the field data the most accurately. The

semi-empirical parameterisation BC83 yields similar results in this altitude range, although it misses the exact values slightly. The Thomsonian theories and the semi-empirical parameterisations show similar results to each other in the troposphere despite having very different temperature dependences in the ternary term. Between 10 and 25 km, Ba85 reproduces the field data RH81 accurately, however, its application is limited because it is only dependent on $h$ and does not contain physical information on the $p$ and $T$ dependencies. Above this altitude range, the picture is more complicated. Field data coverage above

35 km is sparse so that it is difficult to judge the performance of the different theories and parameterisations. Based on the available field data, HF86 shows the best performance, followed by Ar83 and SA82, although both parameterisations come with certain constraints: the ternary term of Ar83 has a weak performance below 25 km and SA82 is solely dependent on $h$ and contains no physical information on $T$ and $p$, similar to Ba85. This inter-comparison ultimately shows that the question of an accurate parameterisation or theory of the recombination coefficient $\alpha$ for the troposphere and stratosphere is not yet solved

and further research is necessary in the future.

### 7.2 Comparison to laboratory data

In a second step, the parameterisations and theories are compared to the constant-pressure (1013.25 hPa assumed) and temperature-dependent (218.15 to 293.15 K) data set reported by Franchin et al. (2015). Here, only theories and parameterisations are used that include the temperature as a parameter and account for the ternary recombination mechanism

(i.e. Ga38, Is57, Na59, Lo60, Ar83, BC83, and HF86). Parameterisations that predict $\alpha$ based solely on the altitude or only describe the binary recombination mechanism are, therefore, excluded (i.e. Fu63, Hi79, SA82, Ba85, and Mi12). We used the same parameters as in the previous sub-section. The result of the inter-comparison is shown in Fig. 3. For Fr15, there is a

general trend towards higher recombination coefficient values for lower temperatures, although the fluctuation is comparably strong. One should bear in mind the possible sources of error from wall losses in the aerosol chamber and sampling line losses of the Neutral cluster and Air Ion Spectrometer (NAIS), the instrument used to determine the mobility distribution of the ions from which the ion-ion recombination rate could be derived. The reported uncertainties for $\alpha$ can be as high as 30 % (Franchin et al., 2015).

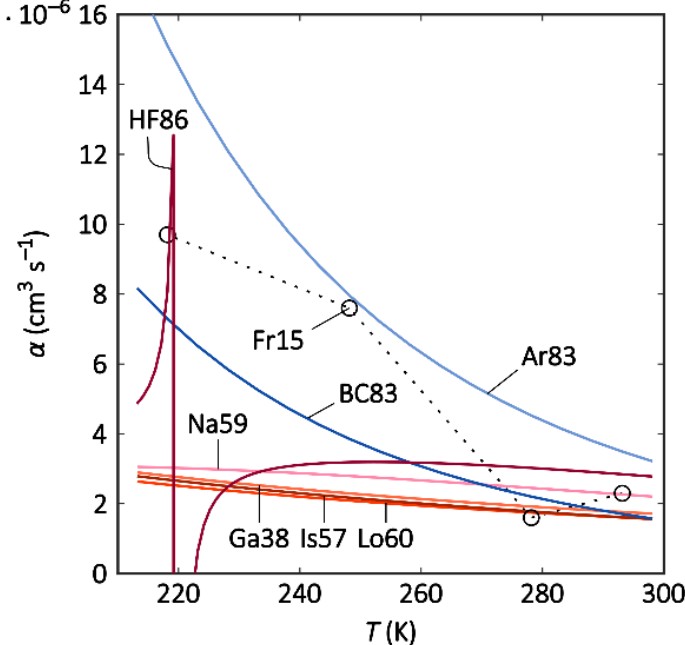

**Figure 3: Parameterisations and theories (solid lines) and laboratory data (circles; dotted lines to guide the eye) of the recombination coefficient $\alpha$ versus $T$ for the temperature range of 213 to 298 K. The meanings of the labels are listed in Table 2.**

All theories and parameterisations agree within a reasonable range at ground level temperatures (270 to 300 K). This is especially the case for the Thomsonian theories Ga38, Is57, Na59, and Lo60; Ga38, Is57, and Lo60 yield almost the same values throughout the considered temperature range. However, the Thomsonian and the semi-empirical theories (Ar83 and BC83) differ tremendously from each other for tropopause temperatures (around 220 K). The weaker temperature dependence of the group Ga38, Is57, Na59, and Lo60 ($T^{-1.5}$) results in a negligible increase of $\alpha$ with decreasing temperatures. The laboratory data of Fr15, on the other hand, show a stronger $T$ dependence (around $T^{-3}$) that is best reproduced by Ar83 and BC83, which assume a $T^{-5}$ dependence for the ternary recombination that is dominant at ground level pressure; BC83 appears to reproduce the data points slightly better than Ar83. HF86 is in the same range as Fr15 for 278 K and 293 K, although it yields much lower values for 248 K compared to Fr15. However, HF86 shows an unexpected behaviour below 230 K, with fluctuating $\alpha$ values that can even become negative. This feature occurs at different temperatures, depending on the input chosen for the ion masses.

In summary, all considered theories and parameterisations can reproduce the laboratory data for warm temperatures (270 and 300 K), while only BC83 and Ar83 can reproduce the $\alpha$ values for colder temperatures (218 and 248 K) because their temperature dependence is more similar to the temperature trend in Fr15. The Thomsonian theories (Ga38, Lo60, Is57, and Na59) only show a weak reaction to reducing temperatures, while HF86 displays an unexpected behaviour within the temperature range considered here.

Franchin et al. (2015) have additionally used the model by López-Yglesias and Flagan (2013) to simulate the ion-ion recombination coefficient for the discussed temperature range. This model describes the ion-aerosol attachment coefficient, although it can also be applied to the special case of two ions recombining. However, the model is unable to reproduce the measured data in the low-temperature regime (Franchin et al., 2015).

## 8 Determination of the ion-ion trapping distance and the collision probability in the limiting sphere

As discussed in the previous section, the ion-ion trapping sphere radius $d$ is an important parameter in the process of the ion-ion recombination. It is connected with the recombination rate $\alpha$ according to Eq. (16) (Natanson, 1959). In order to find the values for the ion-ion trapping radius as a function of pressure and temperature from the measured ion-ion recombination rates (Rosen and Hofmann, 1981; Morita, 1983; Gringel et al., 1978), the equation needs to be solved for $d_N$. Since this cannot be done analytically, it is performed numerically using the Newton-Raphson method. The results are listed in Table B1 and shown in Fig. 4 (a) as an altitude plot, where each data point for $d$ is based on the measured $\alpha$. Only $\alpha$ values above 10 km are considered due to the erroneous determination of the ion-ion recombination coefficient below 10 km as discussed previously. The values for the temperature and pressure are taken from the US Standard Atmosphere (National Oceanic and Atmospheric Administration et al., 1976). The reference values for the electrical mobilities, $\mu_0$, for the conditions of 288.15 K and 1013.25 hPa is $1.3 \cdot 10^{-4}$ m² V⁻¹ s⁻¹ for Gringel et al. (1978) and Morita (1983), whereas $1.5 \cdot 10^{-4}$ m² V⁻¹ s⁻¹ is used for the Rosen and Hofmann (1981) data set. The mobility values, $\mu$, were adjusted for temperature and pressure according to Eq. (32), here with a reference temperature of $T_0 = 288.15$ K. For the calculations it is further assumed that the masses are 90 Da for both the positive and negative ions (see Sec. 7).

All data sets yield similar results for $d$ (see Fig. 4 (a)). The resulting values of $d$ show an increasing trend with altitude; this trend is approximately linear for a logarithmic x-axis. A linear fit of all data points using a logarithmic x-axis yields the altitude-dependent parameterisation given in Eq. (51) and a multivariate fit is performed to determine the $T$ and $p$ dependences (see Eq. (52)):

$$d(h) = 10^{\frac{h-(468\pm15)}{58.5\pm2.0}}. \tag{51}$$

$$d(T,p) = (1.9 \pm 0.3) \cdot 10^{-8} \cdot \left(\frac{T}{T_0}\right)^{1.9\pm0.4} \cdot \left(\frac{p}{p_0}\right)^{-0.19\pm0.02}. \tag{52}$$

It is not possible to extrapolate the parameterisations beyond the input data range of 10 to 45 km. Especially in the troposphere, the temperature trend is opposite to the one in the stratosphere, while the pressure trend is the same. Thus, a conclusion for the

altitude range of 0 to 10 km cannot be drawn from these calculations. Furthermore, the significance of the pressure- and temperature-dependent fit is limited because the temperature only changes by approximately 15 % between 10 and 45 km
altitude which could lead to imprecise results in the temperature dependence.

The gained values for the altitude-dependent Eq. (51) are 15 nm for 10 km, 22 nm for 20 km, and 57 nm for 44 km. The values for the temperature- and pressure-dependent Eq. (52) are similar: 17 nm for 10 km, 21 nm for 20 km, and 59 nm for 44 km. These results are in contrast to the values calculated for Natanson's (1959) theory according to Eq. (17) (see Fig. 4 (a)). Natanson's trapping distance $d_N$ is 15 nm at 0 km altitude, 22 nm at 10 km, and 29 nm at 20 km, reaching a maximum of 30 nm
at 30 km; above this altitude, $d_N$ decreases again, with a value of 27 nm at 44 km. For Hoppel and Frick's (1986) theory, on the other hand, $d_{HF}$ is assumed to be constant at 18 nm. The divergence of the numerical determination presented here, the calculation according to Natanson's formula, and the constant value of Hoppel and Frick highlights that the determination of $d$ is anything but trivial and further research has to be conducted to determine the ion-ion trapping distance for theories that use this parameter in their formulae.

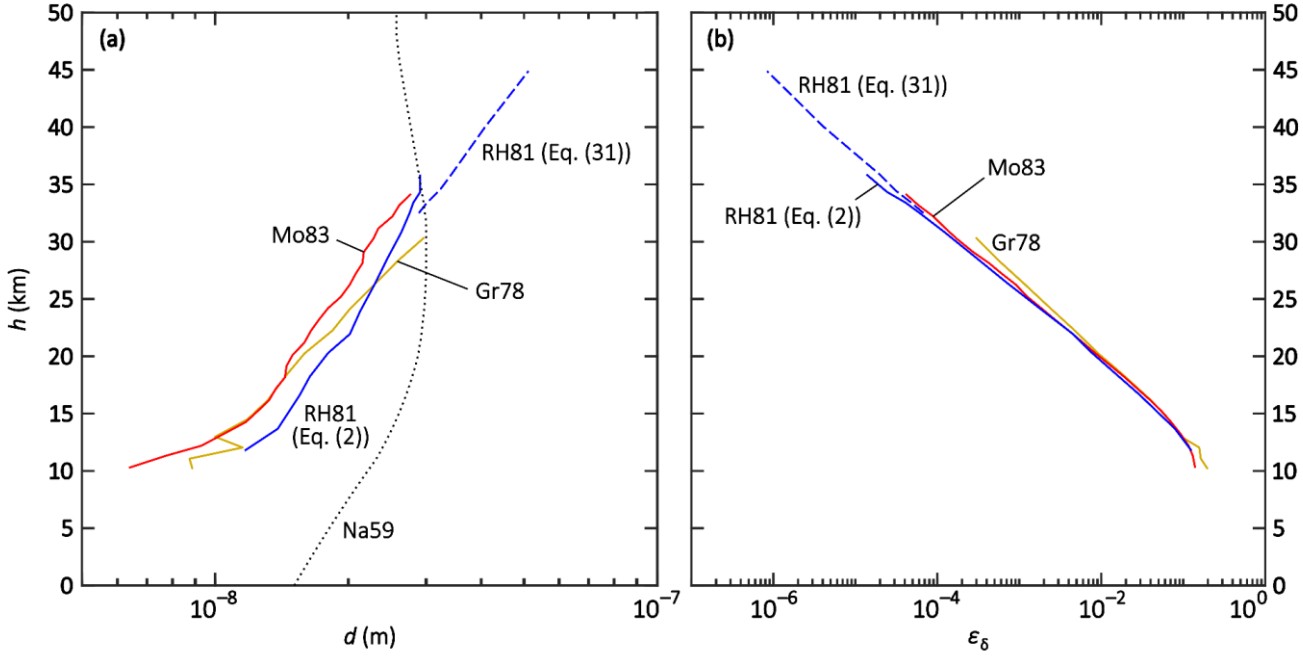


**Figure 4: Altitude plots of a) the numerically determined ion-ion trapping radius $d$ and b) the analytically determined limiting sphere collision probability $\varepsilon_\delta$, each for the field data sets of Gringel et al. (1978) (Gr78, yellow), Rosen and Hofmann (1981) (RH81, blue), and Morita (1983) (Mo83, red). In addition, the dotted curve (Na59) in panel (a) shows $d_N$ after Eq. (17) (Natanson, 1959).**

The formula in Eq. (41) by Filippov (1993) and Tamadate et al. (2020b) to determine the ion-ion recombination coefficient is
generally valid. Other theories, experiments, or models, therefore, only need to determine the collision probability for ions entering the limiting sphere, $\varepsilon_\delta$, in order to be compared with other theories or data sets. To be able to validate the determined values with the field data, $\varepsilon_\delta$ is calculated by analytically solving Eq. (41) for $\varepsilon_\delta$ and using the balloon-borne ion-ion

recombination rates mentioned above as the input variable. The results are listed in Table B1 and are shown in the altitude plot in Fig. 4 (b). All data sets yield similar results and show a decreasing trend for increasing altitudes. With a logarithmic x-axis, this trend is linear and can be described by the parameterisation given in Eq. (53).

$$\varepsilon_\delta(h) = 10^{\frac{h-(7.0\pm0.2)}{-(6.18\pm0.06)}}. \tag{53}$$

For the above-mentioned reasons, this parameterisation is only valid between 10 and 45 km. For instance, $\varepsilon_\delta$ is 0.33 for 10 km altitude, $7.9 \cdot 10^{-3}$ for 20 km and $1.0 \cdot 10^{-6}$ for 44 km. The multivariate fit for $T$ and $p$ does not yield a satisfactory parameterisation for $\varepsilon_\delta$; the deviation from the initially determined values can be as high as one order of magnitude for higher altitudes and, therefore, the $T$ and $p$ dependent parameterisation is omitted.

## 9 Conclusion and outlook

The history of theoretical and empirical approaches to quantify the ion-ion recombination coefficient $\alpha$ and its parameterisations have been reviewed. The parameterisations and theories have been compared to field and laboratory data and to a model calculation of $\alpha$ with a focus on temperature and pressure dependence and their applicability to the troposphere and stratosphere, i.e. from 0 to 50 km altitude. For standard conditions (i.e. 273.15 K, 1013 hPa), the value of $1.7 \cdot 10^{-6}$ cm$^3$ s$^{-1}$ is recommended to be used. Evidence is strong that this value is accurate because several authors have derived it independently from laboratory measurements as well as from model calculations. As of today's knowledge, it is the best assumption to use a nearly constant value for altitudes between 0 and 10 km; this is due to the roughly counterbalancing effects of temperature and pressure on the recombination coefficient. Above 10 km, however, a temperature- and pressure-dependent parameterisation must be used to account for the decreasing value of the ion-ion recombination coefficient. The parameterisation of Israël (1957) shows the best agreement with the field and model data of $\alpha$ for the altitude range of 0 to 22 km. Between 0 and 20 km, the parameterisation by Brasseur and Chatel (1983) also yields good results and, furthermore, it reproduces the recent laboratory measurements the most faithfully. Between 10 and 25 km, the altitude-dependent parameterisation of Bates (1985) reproduces the field data accurately, while for altitudes above 25 km, the parameterisations of Smith and Adams (1982) and Arijs et al. (1983) show the closest resemblance to the field data, although it is difficult to judge this for altitudes above 35 km because of the sparse data coverage above this altitude. Above 30 km altitude, the theory of Hoppel and Frick (1986) shows an excellent agreement with the (sparse) field data.

However, the understanding of the processes in ion-ion recombination is far from complete. Binary and ternary recombination mechanisms both play a role, however, their specific temperature and pressure dependencies are not fully resolved up to the present day. In addition, the ion-ion recombination is influenced by the mobilities and masses of the ions. Moreover, this work only focussed on the recombination in air; additional gases can be investigated in future studies. More experiments and state-of-the-art model simulations, including molecular dynamics simulations, are needed to determine the ion-ion recombination coefficient in dependence of temperature, pressure, ion masses, and ion mobilities. This is crucial in order to calculate accurately the recombination loss of ions for the diverse ambient conditions we observe in our atmosphere.

# Appendix A: Calculation of the collision radius

They collision radius $r_{coll}$ is defined as the sum of the radii of the positive and the negative ion, $r_+ + r_-$, respectively. These can be calculated in dependence of the masses of the two ions, the temperature, and the pressure, according to Eq. (A1) to (A4) (López-Yglesias and Flagan, 2013):

$$r_+ = -r_{gas} + 0.5 \cdot \sqrt{3 \cdot \sqrt{1 + \frac{m_+}{m_{gas}}} \cdot \frac{v_+ \cdot k_B T}{8 \cdot p \cdot D_+}}, \qquad (A1)$$

$$v_+ = \sqrt{\frac{8 \cdot k_B T}{\pi \cdot m_+}}, \qquad (A2)$$

$$r_{gas} = \left( \frac{m_{gas} \cdot k_B T}{16 \pi^2 \cdot \eta^2} \right)^{0.25}, \qquad (A3)$$

$$\eta = \eta_0 \cdot \frac{T_0 + S_C}{T + S_C} \cdot \left( \frac{T}{T_0} \right)^{1.5}, \qquad (A4)$$

where $r_{gas}$ is the radius of the gas molecule, $m_+$ is the mass of the positive ion in kg, $m_{gas}$ is the mass of the gas molecule in kg, $\eta$ is the viscosity of the gas, $\eta_0$ is the viscosity of the gas at standard temperature, and $S_C$ is the Sutherland's constant. $r_-$ can be calculated accordingly by replacing $m_+$ for $m_-$, $v_+$ for $v_-$, and $D_+$ for $D_-$, respectively. $T_0 = 298.15$ K, here, for $m_{gas}$, we assumed 29 Da, for $\eta_0$, we used $1.827 \cdot 10^{-5}$ Pa s (López-Yglesias and Flagan, 2013), and for $S_C$, we used 113 (Chapman and Cowling, 1960).

# Appendix B: Values for the ion-ion trapping distance and the collision probability in the limiting sphere

Table B1: Numerically calculated values for the ion-ion trapping distance, $d$, and analytically calculated values for the collision probability in the limiting sphere, $\varepsilon_\delta$, for the reported field data of the ion-ion recombination coefficient, $\alpha$, taken from Gringel et al. (1978), Rosen and Hofmann (1981), and Morita (1983). The data for $T$ and $p$ are taken from the US Standard Atmosphere (National Oceanic and Atmospheric Administration et al., 1976). For details, see Sect. 8.

| $h$ (km) | $\alpha$ (cm$^3$ s$^{-1}$) | $T$ (K) | $p$ (hPa) | $d$ (m) | $\varepsilon_\delta$ |
|---|---|---|---|---|---|
| *Gringel et al. (1978)* | | | | | |
| 10.2 | 2.14E-06 | 223 | 265 | 1.63E-08 | 1.99E-01 |
| 11.1 | 1.93E-06 | 217 | 227 | 1.60E-08 | 1.64E-01 |
| 12.0 | 2.06E-06 | 217 | 194 | 1.90E-08 | 1.57E-01 |
| 13.0 | 1.42E-06 | 217 | 166 | 1.65E-08 | 9.45E-02 |
| 14.5 | 1.27E-06 | 217 | 131 | 1.83E-08 | 6.64E-02 |
| 16.1 | 1.05E-06 | 217 | 102 | 1.94E-08 | 4.06E-02 |
| 18.4 | 7.82E-07 | 217 | 71.1 | 2.05E-08 | 1.81E-02 |
| 20.3 | 6.37E-07 | 217 | 52.8 | 2.18E-08 | 9.18E-03 |
| 22.3 | 5.73E-07 | 219 | 38.6 | 2.46E-08 | 4.74E-03 |
| 24.1 | 4.95E-07 | 221 | 29.3 | 2.65E-08 | 2.44E-03 |
| 26.3 | 4.44E-07 | 223 | 20.9 | 2.98E-08 | 1.14E-03 |
| 28.3 | 4.06E-07 | 225 | 15.4 | 3.31E-08 | 5.74E-04 |
| 30.4 | 3.92E-07 | 227 | 11.3 | 3.77E-08 | 2.94E-04 |
| *Rosen and Hofmann (1981) (Eq. (2))* | | | | | |
| 11.8 | 1.81E-06 | 217 | 200 | 1.88E-08 | 1.26E-01 |
| 13.7 | 1.54E-06 | 217 | 149 | 2.08E-08 | 7.91E-02 |
| 16.6 | 1.04E-06 | 217 | 94.2 | 2.20E-08 | 2.96E-02 |
| 18.2 | 8.41E-07 | 217 | 73.3 | 2.27E-08 | 1.64E-02 |
| 20.3 | 6.79E-07 | 217 | 52.8 | 2.43E-08 | 7.71E-03 |
| 21.9 | 6.27E-07 | 218 | 41.1 | 2.67E-08 | 4.53E-03 |
| 23.9 | 4.89E-07 | 220 | 30.2 | 2.78E-08 | 1.97E-03 |
| 26.4 | 3.75E-07 | 223 | 20.6 | 2.97E-08 | 7.17E-04 |
| 28.5 | 3.03E-07 | 225 | 15.0 | 3.14E-08 | 3.07E-04 |
| 30.8 | 2.45E-07 | 227 | 10.6 | 3.36E-08 | 1.23E-04 |
| 32.5 | 2.05E-07 | 230 | 8.25 | 3.50E-08 | 6.10E-05 |
| 33.4 | 1.84E-07 | 232 | 7.24 | 3.56E-08 | 4.08E-05 |
| 34.3 | 1.63E-07 | 235 | 6.08 | 3.68E-08 | 2.44E-05 |
| 35.9 | 1.34E-07 | 239 | 5.13 | 3.67E-08 | 1.37E-05 |

| $h$ (km) | $\alpha$ (cm$^3$ s$^{-1}$) | $T$ (K) | $p$ (hPa) | $d$ (m) | $\varepsilon_\delta$ |
|---|---|---|---|---|---|
| *Rosen and Hofmann (1981) (Eq. (31))* | | | | | |
| 32.5 | 2.28E-07 | 230 | 8.25 | 3.66E-08 | 6.78E-05 |
| 33.4 | 2.16E-07 | 232 | 7.24 | 3.81E-08 | 4.79E-05 |
| 34.5 | 2.05E-07 | 235 | 6.08 | 4.06E-08 | 3.07E-05 |
| 35.9 | 1.94E-07 | 239 | 5.13 | 4.30E-08 | 1.98E-05 |
| 40.0 | 1.51E-07 | 250 | 2.87 | 5.10E-08 | 4.14E-06 |
| 44.9 | 1.28E-07 | 264 | 1.53 | 6.35E-08 | 8.37E-07 |
| *Morita (1983)* | | | | | |
| 10.3 | 1.54E-06 | 222 | 255 | 1.28E-08 | 1.40E-01 |
| 11.3 | 1.59E-06 | 217 | 217 | 1.44E-08 | 1.31E-01 |
| 12.2 | 1.57E-06 | 217 | 188 | 1.60E-08 | 1.16E-01 |
| 13.2 | 1.43E-06 | 217 | 161 | 1.70E-08 | 9.27E-02 |
| 14.3 | 1.31E-06 | 217 | 135 | 1.82E-08 | 7.10E-02 |
| 15.2 | 1.18E-06 | 217 | 117 | 1.89E-08 | 5.43E-02 |
| 16.2 | 1.05E-06 | 217 | 100 | 1.95E-08 | 3.95E-02 |
| 17.2 | 9.01E-07 | 217 | 85.8 | 1.98E-08 | 2.74E-02 |
| 18.2 | 7.99E-07 | 217 | 73.3 | 2.03E-08 | 1.94E-02 |
| 19.1 | 6.77E-07 | 217 | 63.7 | 2.03E-08 | 1.32E-02 |
| 20.1 | 5.91E-07 | 217 | 54.4 | 2.06E-08 | 8.98E-03 |
| 21.2 | 5.32E-07 | 218 | 45.8 | 2.16E-08 | 5.99E-03 |
| 22.2 | 4.69E-07 | 219 | 39.2 | 2.21E-08 | 4.00E-03 |
| 23.2 | 4.22E-07 | 220 | 33.6 | 2.29E-08 | 2.70E-03 |
| 24.2 | 3.86E-07 | 221 | 28.8 | 2.38E-08 | 1.85E-03 |
| 25.2 | 3.63E-07 | 222 | 24.4 | 2.52E-08 | 1.26E-03 |
| 26.2 | 3.42E-07 | 223 | 21.2 | 2.62E-08 | 9.08E-04 |
| 27.2 | 3.08E-07 | 224 | 18.2 | 2.70E-08 | 6.07E-04 |
| 28.1 | 2.85E-07 | 225 | 15.9 | 2.78E-08 | 4.28E-04 |
| 29.1 | 2.45E-07 | 226 | 13.7 | 2.79E-08 | 2.73E-04 |
| 30.3 | 2.24E-07 | 227 | 11.5 | 2.92E-08 | 1.73E-04 |
| 31.1 | 2.08E-07 | 228 | 10.2 | 2.99E-08 | 1.26E-04 |
| 32.2 | 2.05E-07 | 229 | 8.63 | 3.21E-08 | 8.91E-05 |
| 33.2 | 1.87E-07 | 232 | 7.45 | 3.32E-08 | 5.86E-05 |
| 34.2 | 1.81E-07 | 234 | 6.45 | 3.52E-08 | 4.10E-05 |

# Appendix C: Nomenclature

$b$      critical collision parameter, in m

$d$      ion-ion trapping distance or trapping sphere radius, in m

$d_{ia}$      three-body trapping distance in ion-aerosol attachment (after Hoppel and Frick), in m

$d_{HF,N}$      ion-ion trapping distance (after Hoppel and Frick, Natanson), in m

| | $d_T$ | ion-ion trapping distance, radius of the collision sphere around each ion, or radius of mutual Coulomb attraction between two ions of opposite charge (after Thomson), in m |
|---|---|---|

$d_T$      ion-ion trapping distance, radius of the collision sphere around each ion, or radius of mutual Coulomb attraction between two ions of opposite charge (after Thomson), in m

$D$        diffusion coefficient, sum of $D_+$ and $D_-$, in $m^2\ s^{-1}$

$D_{+,-}$      diffusion coefficient of the positive, negative ion, in $m^2\ s^{-1}$

$D_{+/-,0}$      diffusion coefficient of the positive/negative ion at standard temperature and pressure, in $m^2\ s^{-1}$

$D_{ion}$      diffusion coefficient of one ion, in $m^2\ s^{-1}$

$e$        electron charge, $1.602\ 177 \cdot 10^{-19}$ C

$E$        external electrical field, in $V\ m^{-1}$

$EA$      electron affinity, in eV

$h$        altitude, in km

$k_B$      Boltzmann constant, $1.380\ 649 \cdot 10^{-23}\ J\ K^{-1}$

$Kn_\delta$      Knudsen number for the limiting sphere

$L$        loss rate to the electrodes, in $cm^{-3}\ s^{-1}$

$[M]$      number density of air molecules, in $cm^{-3}$

$m_{+,-}$      mass of the positive, negative ion, in Da (unless noted otherwise)

$m_{gas}$      molecular mass of the gas, in kg

$m_{ion}$      ion mass, in Da (unless noted otherwise)

$m_{red}$      reduced mass, in kg (unless noted otherwise)

$n$        number concentration of ions in the gas phase, in $cm^{-3}$

$n_{+,-}$      number concentration of the positive, negative ions in the gas phase, in $cm^{-3}$

$n_{total}$      number concentration of the sum of negative and positive ions in the gas phase, in $cm^{-3}$

$p$        pressure, in hPa

$p_0$      standard pressure, 1013.25 hPa

$q$        ion pair production rate, in $cm^{-3}\ s^{-1}$

$r$        distance of the two ions, in m

$r_0$      initial distance of the two ions, in m

$r_{coll}$      collision radius, sum of the radii of the positive and negative ions, in m

$r_{gas}$      radius of the gas molecule, in m

$r_p$      particle radius, in m

$RH$      relative humidity, in %

$S_C$      Sutherland's constant, 113 for air

$t$        time, in s

$T$        temperature, in K

$T_0$      standard temperature, 273.15 K

| | | |
|---|---|---|
| | $v$ | equilibrium ion drift velocity, in m s$^{-1}$ |
| | $v_{+,-}$ | mean thermal speed of the positive, negative ion, in m s$^{-1}$ |
| | $v_{rel}$ | relative thermal speed of two ions, in m s$^{-1}$ |
| 795 | $w_{T,N}$ | function of $x$ (used in Thomson theory and Natanson's theory) |
| | $x$ | function of $d_T$ and $\lambda_{ion}$ (used in Thomson theory) |
| | $x'$ | function of $T$ and $p$ (used in Thomson theory) |
| | $x_N$ | function of $d_N$ and $\lambda_{ion}$ (used in Natanson's theory) |
| | $\alpha$ | ion-ion recombination coefficient, in cm$^3$ s$^{-1}$ |
| 800 | $\alpha_2$ | binary ion-ion recombination coefficient, in cm$^3$ s$^{-1}$ |
| | $\alpha_3$ | ternary ion-ion recombination coefficient, in cm$^3$ s$^{-1}$ |
| | $\alpha_\delta$ | ion-ion collision rate coefficient at distance $\delta$ (limiting sphere surface), in m$^3$ s$^{-1}$ |
| | $\gamma$ | function of $r_{coll}$ or $d_{HF}$ (used for Fuchs's and Hoppel and Frick's theories) |
| | $\delta$ | limiting sphere radius, in m |
| 805 | $\delta_F$ | Fuchs's limiting sphere radius, in m |
| | $\delta_{HF}$ | Hoppel and Frick's limiting sphere radius, in m |
| | $\Delta\alpha_2$ | enhancement to the ion-ion recombination coefficient due to the binary channel, in cm$^3$ s$^{-1}$ |
| | $\varepsilon_0$ | vacuum permittivity, $8.854\,188 \cdot 10^{-12}$ A s V$^{-1}$ m$^{-1}$ |
| | $\varepsilon_d$ | probability factor for ions entering the trapping sphere |
| 810 | $\varepsilon_{L,T}$ | ratio of successful recombinations per collision (after Langevin, Thomson) |
| | $\varepsilon_N$ | ion-gas molecule collision probability, also named "adsorption coefficient" (after Natanson) |
| | $\varepsilon_\delta$ | collision probability for ions entering the limiting sphere |
| | $\eta$ | viscosity of a gas, in Pa s |
| | $\eta_0$ | viscosity of air at 298.15 K, $1.827 \cdot 10^{-5}$ Pa s |
| 815 | $\theta$ | critical angle to enter the trapping sphere |
| | $\lambda$ | ion-ion mean free path, in m |
| | $\lambda_{+,-}$ | mean free path of the positive, negative ion, in m |
| | $\lambda_{air}$ | mean free path of air, in m |
| | $\lambda_{ion}$ | mean free path of one ion, in m |
| 820 | $\lambda'$ | mean distance of the last collision of a particle with a gas molecule before striking the limiting sphere surface (after Fuchs), in m |
| | $\mu$ | ion mobility, in m$^2$ V$^{-1}$ s$^{-1}$ |
| | $\mu_{+,-}$ | ion mobility of the positive, negative ion, in m$^2$ V$^{-1}$ s$^{-1}$ |
| | $\mu_0$ | ion mobility at standard temperature and pressure, in m$^2$ V$^{-1}$ s$^{-1}$ |
| 825 | $\sigma$ | electrical conductivity of the air, in S m$^{-1}$ |

$\Psi_\delta$       ratio of Coulomb and thermal energy at distance $\delta$ (limiting sphere surface)

## Author contribution

MZW designed the study, surveyed the literature and compared the parameterisations, models and data sets. MZW and AK modelled the parameterisations and theories. MZW, AK, and JC discussed the results. MZW and AK wrote the manuscript; JC provided input for revision before submission.

## Competing interests

The authors declare that they have no conflict of interest.

## Acknowledgements

*tbw*

## Financial support

Marcel Zauner-Wieczorek is funded by the Heinrich Böll Foundation.

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
