# Peer review of "The ion-ion recombination coefficient $\alpha$ : Comparison of temperatureand pressure-dependent parameterisations for the troposphere and stratosphere"

_Atmospheric Chemistry and Physics, 2021_

## Author Response (AR1)

Authors' response to the Referee Comments 1–3

Comments by the Referees are in black, answers by the authors are in blue, citations from the manuscript are in light blue

**Answers to Referee 1**

This manuscript reviews and then attempts to apply various theories predicting the ion-ion recombination rate in comparison to select experimental data. I think this is certainly a topic worthy of study. However, my recommendation is between major revision and rejection, because I believe this work is misguided in its approach and has more than several inaccurate statements in it. Much of this has to do with the manner in which the authors compare to "Ta20" (which is really a comparison to Fuchs 1963, not Ta20), and I do not believe the "intercomparison" of theories approach with subjective choices in inputs is a reasonable way to go about scientific study. Ultimately, I would like to think the authors can improve upon this work, and endorse major revision.

We would like to thank the referee for their feedback. We believe that thanks to the feedback provided by this referee and the other two referees, severe misconceptions and errors were eliminated, chapters and paragraphs were rearranged to ensure a better comprehensibility, and new ideas were implemented to improve the quality of the manuscript.

Many theories and parameterisations exist to determine the ion-ion recombination coefficient. They yield different results and it is not possible to conclude on one particular theory or parameterisation that covers the whole troposphere and stratosphere, as our study shows. Therefore, we are convinced that it is necessary to discuss and compare those different approaches to better understand their validity/applicability range and to identify further research that is needed.

Comments:

1. As noted in the prior paragraph, I believe the article is quite misguided in its approach. I think this is most apparent in the works presentation and discussion of Tamadate et al (2020). The article devotes quite a bit of time discussing the work of Tamadate et al (2020), and in fact in looking at Tamadate et al 2020, it looks like a considerable fraction of the review section of this article is based upon the introduction of Tamadate et al. Specifically, a large fraction of the references reviewed in this work are similarly discussed in Tamadate et al, and the symbols and notation in the present manuscript also appear to be taken directly from Tamadate et al in the number of instances (this manuscript even refer readers to Tamadate et al 2020 for Hoppel and Frick's equations, instead of referring readers to Hoppel and Frick!). However, in reading Tamadate et al (2020), I come away thinking the authors completely miss the purpose of that article (or for some reason, do not want others to consider developing and expanding the Tamadate et al 2020 approach). Tamadate et al (2020), and the same researchers subsequent study (doi: 10.1039/D0CP03989F) focus explicitly on leveraging Molecular Dynamics simulations to determine the collision probability needed for implement in Filippov's generalized version of the limiting sphere theory. When the authors attempt to compare their prior measurements to Tamadate et al (2020), they state:

"We calculated Ta20 based on the derivations after Fuchs (1963) (Eq. (34) to (36)) and after Filippov (1993) (Eq. (37) to (39)), using Eq. (40) to (42) likewise. However, both derivations yielded the same results within our limits of uncertainty, therefore, for a better overview, for Ta20 we only show the results based on Fuchs."

Using the relationship of Fuchs 1963 with the limiting sphere of Wright is nothing more than Fuchs original theory, and not a test of what Tamadate et al did (equations 34-36 are Fuchs's exact theory, Tamadate et al just reiterates them). To be clear, Tamadate et al did not derive new equations, they implement Filippov's equations with Molecular Dynamics simulations, and without using results from MD simulations specific to the ions, gas composition, temperature, and pressure of interest, comparison is not being made appropriate to their work. The simulations in Tamadate et al (2020) agree with Fuchs when they neglect gas molecule-ion interactions (validating their approach), but this is not intended to be an accurate calculation of the recombination rate. Their simulations lead to much higher recombination rates than those of Fuchs and would lead to different values than the predictions here. I suggest correcting this comparison to note it is a comparison to Fuchs's theory, not to Tamadate et al's hybrid continuum-MD simulation approach. Disagreement between measurement and Fuchs's approach when applied to the ion-ion recombination has been known for decades.

In addition to the incorrect comparison, the statements about Tamadate et al are also largely inaccurate:

" Thus, they restricted the MD simulation to the limiting sphere while using the continuum (diffusion) equations outside the limiting sphere…." They actually use a cubic simulation domain of gas molecules that follows both ions. Simulations do not necessarily use Fuchs's definition of the limiting sphere, but they adjust the sphere radius used as the boundary between continuum and MD to ensure that this radius is large enough not to influence results.

"The MD simulations were run for different conditions: with and without the influence of electrostatic forces," Tamadate et al (2020) do not run simulations excluding electrostatic forces (which are extremely important in this problem). They do appear to include and exclude the initial electrostatic velocity for the incoming ions in the limiting sphere theory, but this is very different from including or excluding forces.

"In order to derive the recombination coefficient, they used two different approaches: the theory by Fuchs (1963) and the one by Filippov (1993)." Filippov's 1993 approach is a more general version of Fuchs 1963 (and earlier) derivation. They are not different approaches. Tamadate et al (2020) very clearly uses Filippov's equations and states this unambiguously. Tamadate et al do retrace the steps of Fuchs and Filippov, but I believe they appropriately credit where these steps come from.

"In Fig. 5 (e), the limiting sphere theories Na59, HF86, and Ta20 are shown. Whilst Na59 and HF86 agree fairly well with each other, Ta20 yields α values which are one order of magnitude too low (2.7 · $10^{-6}$ cm$^3$ s $^{-1}$ at ground level) and is, therefore, not recommended." To reiterate, the plots displayed are not an accurate test of Ta20, as the probability of Fuchs was used. This statement is hence very inconsistent with the earlier statement in this manuscript, "While the approach of Tamadate et al. (2020) is very promising, they correctly emphasise the need for hybrid continuum-MD simulations with N2 and O2, instead of He, in order to achieve results comparable to atmospheric conditions." The authors here have not made the appropriate comparison.

We are very thankful to the referee to pointing out these misconceptions of ours. We revised the manuscript thoroughly with regards to the description of the work of Tamadate et al. (2020) and the works by Fuchs (1963) and Filippov (1993), following the referee's advice and rectifications. For the comparison of theories, we used Fuchs's (1963) theory and named it accordingly. The description of the work of Tamadate et al. (2020) was completely rewritten and put into a new chapter on numerical model simulations (Section 6). Moreover, we revised the chapter on the theories of Fuchs (1963) and Hoppel and Frick (1986), clearly pointing to the introductory part of Tamadate et al. (2020). We believe it is important to add these theories to the discussion and overview, while also giving proper credit to the work done by Tamadate et al. Where applicable, we used the formulae of the original theories; when the formulae developed by Tamadate et. al appeared to be more convenient to use, we

introduced these to the reader, while citing Tamadate et al. Furthermore, we revised the symbols and notations used in the manuscript to ensure consistency.

I also believe the authors are mistaken in the computational power and expertise required to perform such MD calculations. Certainly MD approaches need to be developed further to make use easier. However, it is not unfeasible to use MD calculations to compute and tabulate the ion-ion recombination rate under a variety of conditions. I do not agree with the statement " Simulation experiments at temperatures and pressures representative of the different layers of the lower atmosphere could provide a better insight into the variation of the ion-ion recombination coefficient $\alpha$ in the atmosphere. Eventually, parameterisations are needed for everyday us because MD simulations require advanced computing power and experience." The MD simulation approach the authors are discussing is only ~1 year old, and notion that this cannot become a common approach to compare to data, or even to predict the recombination rate in the future seems short-sighted and overly dismissive.

We apologise for this misunderstanding. It was never our intention to discredit MD simulations or claim that their area of application cannot be expanded in the future. To avoid any misunderstandings, we did not add further reflections on the complexity of the usage of MD simulations in the newly written chapter about numerical simulations.

2. Second, the comparison to Hoppel & Frick (1986) is odd. Hoppel & Frick specifically developed a theory to describe the ionisation of particles, and use the ion-ion recombination coefficient as an input to bracket results (their concern was ensuring that the rate of small particle-ion recombination agreed with the ion-ion recombination rate and noticed that in Fuchs's theory this would not be the case, so they worked rather hard to develop an approach taking the essence of Fuchs's theory but which would converge to the ion-ion recombination rate). Stated differently, they use the ion-ion recombination rate as an input to their theory, not an output. To quote Hoppel & Frick: "The value of the recombination coefficient for atmospheric ions is here taken to be that given by Nolan (1943) as 1.4 x $10^{-6}$ cm$^3$ s$^{-1}$. For any value of ionic mass, a corresponding value of the ion-ion trapping distance d can be determined." If the authors choose to compare to Hoppel and Frick, then I believe they should make clear for each comparison what the reference recombination rate being used is and what the temperature and pressure is for it- did they use the same as Hoppel and Frick of 4 x 10^-6 cm^3 s^-1 at atmospheric pressure and room temperature?

Indeed, Hoppel and Frick's (1986) theory is concerned with the ionisation of particles. But since it predicts the ion-aerosol attachment so well, it appeared interesting to us to apply this theory to ion-ion recombination, which can be considered as a special case of the ion-aerosol attachment (assuming the "aerosol particle" has the size of an ion and is singly charged).

We re-wrote the description of Hoppel and Frick's (1986) work to accurately account for the purpose of their theory (now in Sect. 5).

We stated the input value of $1.7 \cdot 10^{-6}$ cm$^3$ s$^{-1}$ to retrieve the trapping distance $d$ from Fig. 3 in Hoppel and Frick (1986) in the old manuscript (ll. 484ff.) as well as in the new manuscript (Sect. 7.1, first paragraph). Due to the new arrangement of the chapters, we hope that this information becomes more prominent now. Now, all the input values are listed in one paragraph before discussing the results of the comparisons.

3.  Based on comments 1 and 2, I do not agree with the "intercomparison" approach- this treats various theories as fixed and isolated approaches from one another, as opposed to bodies of work building off one another.  Rather than perform an intercomparison of different theories where inputs are selected in advance and the theory is determined to be applicable to the data or not, I believe a healthier approach would be to use the data presented to determine the most ambiguous parameters in theories.  For example, in the case of limiting sphere theories, the most appropriate thing to do would be to determine p($\delta$) in equation (37), the probability needed to find agreement with experimental data.  This would be much more useful than an intercomparison, and would enable the authors to discuss how this probability varies.  Similarly, the authors can determine the value of "d" needed in equation (26) for agreement with data.  I believe Tables of p($\delta$) and d for different temperatures, pressures, and relative humidities would be quite useful and referred to extensively by others.  I strongly suggest the authors to adjust their approach to provide such tables, as opposed to an intercomparison approach which is skewed by subjective choices in inputs.

We agree that the choice of input values for the different theories does not come without trouble. Indeed, we want to draw attention to this circumstance and argue to take caution when choosing the input values. However, the theories and parameterisations have not been developed without reason. Some of them were explicitly developed to determine the ion-ion recombination coefficient for different altitude or temperature and pressure regimes. In fact, there is a need to determine concrete values of $\alpha$ for different altitudes; it was our initial motivation to review all the theories on ion-ion recombination because we needed accurate values of $\alpha$ in different altitudes to be able to analyse airborne field data (see Zauner-Wieczorek et al., 2022). To review and compare the available parameterisations and theories to determine these values, thus, appears to be worthwhile and necessary to us.

We stuck to the specifications and input values given in the theories' original works (such as the trapping distance in Hoppel and Frick's article) because we believe that the theories should be evaluated holistically. We, therefore, do not share the referee's view that the input parameters are subjective choices. Of course, suggestions can and should be made how to improve certain parameters. Therefore, and also based on the suggestion to determine values for $d$ and $p_\delta$ (now called $\varepsilon_\delta$ in the manuscript) mentioned above by the referee, we introduced a new chapter (Sect. 8) in which the corresponding values for $d$ and $\varepsilon_\delta$ were determined for the field data of $\alpha$ reported by Gringel et al. (1978), Rosen and Hofmann (1981), and Morita (1983). The resulting values for $d$ and $p_\delta$ are shown in the new Fig. 4 and are listed in Table A1, while we omitted the previous chapter on the sensitivity study of $d$ for Natanson's (1959) and Hoppel and Frick's (1986) theories.

4.  I would also encourage the authors to expand the data set they use in comparison. There is no reason to limit to atmospheric air when comparing theories.

We agree that it would be very interesting to compare the different theories with respect to different gases. However, the scope of our work is the applicability to the atmosphere and, thus, we limited ourselves to atmospheric air. Nevertheless, we want to encourage other researchers to expand the comparison to other systems and, thus, added this sentence to the conclusion:

p. 27, l. 273: "Moreover, this work only focussed on the recombination in air; additional gases can be investigated in future studies."

5.  The authors do neglect the recent equations of Chahl & Gopalakrishnan (doi: 10.1080/02786826.2019.1614522) who focused on small nanoparticle-ion collisions, but their equations could be extended to ion-ion recombination easily.

The work of Chahl and Gopalakrishnan (2019) is indeed very exciting. In our opinion, its application to ion-ion recombination would justify an article on its own and, thus, we think that it cannot be dealt with appropriately within the limited space in this review article.

Editorial Comments:

1. The line colors in most plots are too similar to one another, and I have a tough time linking the lines in plots to the legend.

We introduced labels next to the lines within the plots and omitted a separate legend to increase the comprehensibility.

References

Chahl, H. S. and Gopalakrishnan, R.: High potential, near free molecular regime Coulombic collisions in aerosols and dusty plasmas, Aerosol Science and Technology, 53, 933–957, https://doi.org/10.1080/02786826.2019.1614522, 2019.

Filippov, A. V.: Charging of Aerool in the Transition Regime, Journal of Aerosol Science, 24, 423–436, https://doi.org/10.1016/0021-8502(93)90029-9, 1993.

Fuchs, N. A.: On the stationary charge distribution on aerosol particles in a bipolar ionic atmosphere, Geofisica pura e applicata, 56, 185–193, https://doi.org/10.1007/BF01993343, 1963.

Gringel, W., Käselau, K. H., and Mühleisen, R.: Recombination rates of small ions and their attachment to aerosol particles, Pure and Applied Geophysics, 116, 1101–1113, https://doi.org/10.1007/BF00874674, 1978.

Hoppel, W. A. and Frick, G. M.: Ion—Aerosol Attachment Coefficients and the Steady-State Charge Distribution on Aerosols in a Bipolar Ion Environment, Aerosol Science and Technology, 5, 1–21, https://doi.org/10.1080/02786828608959073, 1986.

Morita, Y.: Recent measurements of electrical conductivity and ion pair production rate, and the ion-ion recombination coefficient derived from them in the lower stratosphere, Journal of Geomagnetism and Geoelectricity, 35, 29–38, https://doi.org/10.5636/jgg.35.29, 1983.

Natanson, G. L.: The Theory of Volume Recombination of Ions, Journal of Technical Physics, 4, 1263–1269, 1959.

Rosen, J. M. and Hofmann, D. J.: Balloon-borne measurements of electrical conductivity, mobility, and the recombination coefficient, Journal of Geophysical Research: Oceans, 86, 7406–7410, https://doi.org/10.1029/JC086iC08p07406, 1981.

Tamadate, T., Higashi, H., Seto, T., and Hogan, C. J., J.: Calculation of the ion-ion recombination rate coefficient via a hybrid continuum-molecular dynamics approach, The Journal of Chemical Physics, 152, 94306, https://doi.org/10.1063/1.5144772, 2020.

Zauner-Wieczorek, M., Heinritzi, M., Granzin, M., Keber, T., Kürten, A., Kaiser, K., Schneider, J., and Curtius, J.: Mass spectrometric measurements of ambient ions and estimation of gaseous

sulfuric acid in the free troposphere and lowermost stratosphere during the CAFE-EU/BLUESKY campaign, Atmos. Chem. Phys. Discuss., https://doi.org/10.5194/acp-2022-238, 2022.

**Answers to Referee 2**

The paper by Zauner-Wieczorek, Curtius, and Kueten discusses the history of ion-ion neutralization measurements and theory in the atmosphere. The authors have done a good job of digging up many old references, some of which are new to me and my group although we work in this field. However, I and other members of my group have some serious issues with the paper. I wonder if all this detail on the chemical physics of the process is completely lost on atmospheric chemists.

We would like to thank the referee for their feedback. Based on the feedback provided by this referee and the other two referees, we revised the manuscript, eliminated misconceptions and restructured the manuscript for a better readability.

The manuscript is very hard to read with lots of jargon, many references to various studies that are hard to keep in ones mind.

We have carefully re-organised the paper with the intention to improve the structuring and therefore also the flow and readability of the paper. The revised version was also edited by a native speaker and we have tried to reduce the use of jargon. If the reviewer still has suggestions about the use of jargon or passages that are hard to read then we would be thankful if the specific passages or phrases could be pointed out by the reviewer.

The manuscript is full of confusion between total recombination rate constants without separating what refers to two body and three body contributions are added. I would start with the simpler story of two body recombination and start adding three body processes in the introduction. I realize that some of the early work measures the total rate but going back and forth is difficult so that one compares things that shouldn't be compared.

We followed the referee's suggestion to highlight the differences between binary and ternary recombination more. Thus, we created a separate chapter for the binary theories (Sect. 3). We agree that commencing with the binary process and then introducing the ternary process would be a possible approach. Because the theories build up on each other, however, we found it more comprehensible for the readers if the theories are introduced more or less chronologically, but with a clear separation of the binary theories. We uniformly introduced the index "2" to refer to binary theories ($\alpha_2$), including Table 2. We kept the index-less $\alpha$ for the total recombination coefficient, which most theories and parameterisations in the manuscript describe. Moreover, we expanded the altitude range of the comparison from 0–20 km to 0–50 km; this way, a comparison of the binary theories as well as the total recombination theories in the upper altitude range (where the two-body recombination predominates) is possible. Furthermore, we added some basic information on the binary/ternary processes in the introductory chapter (Sect. 1) to guide the readers from the beginning:

p. 2, l. 38ff.: "There are two important recombination mechanisms: binary recombination, in which two ions of opposite sign recombine upon collision, and ternary recombination, in which one of the ions first has to collide with a neutral gas molecule, i.e. the third body, to dissipate energy in order to recombine successfully with the second ion. The latter process is, therefore, also called three-body trapping. When both the binary and ternary processes are included in a theory or parameterisation, it is called total recombination."

The early history of neutralization is interesting and worth noting, but in my opinion not worth all the detail and equations given in the manuscript. While early researchers like Thomson knew that ion-molecule reactions apparently take place, actual rate constants weren't measured until the 1950s and 1960s, meaning that early researchers couldn't appreciate the complexity of the ion types that were actually involved in the neutralization. It's my opinion that the quantitative similarity of early measurements is mostly a coincidence. Even Loeb in his later books said that it was only after WWII that electronics were advanced sufficiently to make decent measurements. (This from memory; it would take some time to find exactly what Loeb said.)

It is difficult for us to judge if it is a coincidence when the early findings agree with today's findings. Loeb's statement can be found in his textbook that we already cited (Loeb, 1960). We believe it is important to explain the first steps in theory and experiments because the later development is based on those beginnings. However, we added a caveat to our manuscript to highlight this issue so that readers may evaluate the early results cautiously:

p. 4, l. 89ff.: "Moreover, Loeb (1960) pointed out that measurement techniques were not sophisticated enough and gases not pure enough before the 1950s to be able to determine accurately the ion-ion recombination."

p. 9, l. 229ff.: "Furthermore, one has to bear in mind that the capabilities of the instruments and the purity of the gases were less advanced before the 1950s (Loeb, 1960). Therefore, results obtained before that time need to be considered with caution."

The manuscript is mainly concerned with 3-body neutralization, but it seems to me that binary and ternary measurements or theory are not well distinguished, for example, they are mixed in Table 2. An example I know something about: both the Hickman (incorrectly evaluated) and Miller expressions are plotted vs altitude in Fig. 4(a) even though both are solely for binary neutralization and completely inappropriate for a plot vs altitude.

As mentioned above, we made a clearer distinction between total, ternary, and binary recombination and added the index "2" ($\alpha_2$) for the purely binary theories. By expanding the altitude plot (former Fig. 4, now Fig. 2) to the altitude range of 0 to 50 km, the discussion of the binary theories becomes relevant for the upper part of the stratosphere. We only show the binary theories above 40 km altitude. Furthermore, we omitted the binary theories from Fig. 3, where only ternary theories apply.

The 3 body work of Smith and Adams has been questioned by Rainer Johnsen.

We added this information and reference where we first present the parameterisation by Smith and Adams (1982):

p. 11, l. 286: "Furthermore, Smith and Adams (1982) presented a parameterisation valid for the altitude range of 10 to 60 km based on laboratory measurements of binary recombination with the flowing afterglow/Langmuir probe (FALP) technique. The resulting parameterisation is simple because it only depends on the altitude and contains two terms that represent the ternary and binary recombination, respectively, as Eq. (30) shows:

$$\alpha = 1.63 \cdot 10^{-5} \cdot e^{-\frac{h}{7.38}} + 5.25 \cdot 10^{-8}, \tag{30}$$

where *h* is the altitude in km. The two terms of Eq. (30) represent the ternary and binary recombination, respectively. Johnsen et al. (1994) later disputed their results because they found that ion-collecting probes, as used by Smith and Adams, are not suitable to obtain data on ion-ion recombination coefficients in flowing-afterglow studies."

Beginning with line 255, the results of Hickman are quoted incorrectly.  The formula in Eq. (24) is Hickman's, however, Zauner-Wieczorek et al. says that Hickman's reduced mass is in amu, but that's not right; Hickman used reduced masses in atomic units (the mass of the electron).  Use of the formula as stated by Zauner-Wieczorek et al. would lead to rate constants 200 times too large.

Further, some particular data are incorrectly quoted from Hickman's paper.   It's important to note that those data were not Hickman's.  He was using data from the SRI merged beams experiments.  It is now known that the SRI molecular ions were highly vibrationally excited (if not electronically excited), as was later shown by the SRI people themselves with a new collinear ion-laser experiment, which is the reason no further measurements were made with their merged beams apparatus.  The important point is that the data quoted are incorrect because Zauner-Wieczorek et al. assumed that the units were E-06 cc/s, but Hickman clearly states that the units are given in Fig. 4, where E-08 cc/s is stated.  The same units are specified in Fig. 3 along with the units for m (atomic units).

Beginning with line 260, the results of Miller are quoted incorrectly.  The quoted formula is the same as Hickman's except that the reduced mass is given in amu instead of Hickman's atomic units.  So the formula should not be attributed to Miller.  Miller used flowing afterglow data that existed at that time (1979) to improve on Hickman's parameterization instead of using the faulty SRI merged beams data.  The formula developed by Miller is not quoted by Zauner-Wieczorek et al., namely, $a = 3.32E\text{-}07 \ (T/300)^{-0.5} \ m^{-0.52} \ EA^{-0.24}$.  The "T<1000K" is a limit imposed because the neutralization cross section is known to depend on 1/E at least for such temperatures.

The results of Hickman and Miller are consequently misstated in Table 2, and even worse in Fig. 3, where Hickman's rate constants lie two orders of magnitude above Miller's.  Surely this discrepancy should have tipped off one of the authors to reexamine those two papers.  Fig. 4 is likewise misleading.

The 1980 paper of Miller is only of historical interest and shouldn't be considered in this manuscript at all.  The type of analysis attempted by Miller in 1980 has been superseded by a more recent paper utilizing far more data: T. M. Miller, N. S. Shuman, and A. A. Viggiano, "Behavior of rate coefficients for ion-ion mutual neutralization, 300-550 K" J. Chem. Phys. 136, 204306 (2012), in which these parameterizations were given (m in amu, EA in eV):

$a = (2.8 \pm 1.0)E\text{-}07$ cc/s $(T/300)^{-0.9\pm0.1} \ m^{-0.5\pm0.1} \ EA^{-0.13\pm0.04}$ for polyatomic ions

$a = (3.2 \pm 1.4)E\text{-}08$ cc/s $(T/300)^{-1.1\pm0.2} \ m^{-0.01\pm0.09} \ EA^{-0.04\pm0.23}$ for diatomic ions.

We are thankful to the referee to point out the misunderstanding and misinterpretation in the use of the units of Hickman's (1979) formula and the essential background information regarding Miller's (1980) data. The concept of the Hartree atomic unit (a.u.), i.e. the relative mass based on the mass of the electron, was indeed misinterpreted as atomic mass unit (a.m.u.). We now changed the units of Hickman's formula. We had not made use of the new formula presented in Miller (1980) because it was given for $T \geq 1000$ K. In the article of Miller et al. (2012), we found a footnote that this was a misprint and it should read $T \leq 1000$ K.

We follow the referee's suggestion and do not consider Miller (1980) any more, but include the work by Miller et al. (2012). We changed the text in the main body accordingly.

We also modified Fig. 4 (now Fig. 2), which now spans an altitude range of 0 to 50 km, to account for correct values for Hickman (1979) and to include the more recent parameterisation by Miller et al. (2012) for polyatomic ions above 40 km altitude because in the upper stratosphere, the binary recombination process is predominant and it is, thus, interesting to compare binary theories.

Besides the more recent work of the AFRL group using the VENDAMS technique to derive the above parameterizations, they also miss exciting new work from the Urbain Group and DESIREE group in Stockholm. Also the Prague group has done the most fundamental work on three body increases to the overall rate constants. No mention of product formation is mentioned. Not always is the process a simple electron transfer.

Apart from Miller et al. 2012, we also included references to Shuman et al. (2014a), Shuman et al. (2014b), and Wiens et al. (2015) from the AFRL group who extended the dataset to experiments with several other ions. We believe that the referee means Smith and Španěl when mentioning the Prague group. By citing Smith's earlier works (Smith and Church, 1977; Smith and Adams, 1982), we believe that we ascribe credit to his fundamental contribution to this field of research. We also reviewed the publications by the Urbain group (UC Louvain) and the DESIREE Group in Stockholm. While their work is extremely fascinating, its scope of application – as they state themselves – are astrophysical or stellar systems. The sub-thermal conditions they studied are beyond the scope of this paper that is concerned with the applicability of theories and parameterisation to the Earth's lower atmosphere. To guide the readers to more information on the product formation and the processes other than electron transfer, we added the following sentences to the manuscript:

p. 10, l. 267ff.: "While most research in the field has been carried out on the recombination process itself, some works have also been devoted to study the product formation. For instance, Shuman et al. (2010) investigated the different product channels of the recombination of $SF_{4-6}^-$ with $Ar^+$. Besides simple electron transfer reactions, the elimination of F atoms was also observed."

The electron affinity of NO3- is 4 eV not 1eV.

We changed the value accordingly to 3.94 eV (Weaver et al., 1991).

Why is H3O+(H2O)3 represented by mass 150? I know the answer is other positive ions exist but that should be clear.

In the revised version of the manuscript, we decided to use the mass of 90 Da for positive and negative ions and the related values according to López-Yglesias and Flagan (2013) to ensure uniformity in the calculations throughout the manuscript.

Given that the work we know well is misrepresented, we, of course, worry that more work has also been misrepresented.

This paper needs at least a major rewrite and I don't believe it belongs in this journal.

Based on the feedback by this referee and the other two referees, we included major changes to our manuscript to remove errors and misinterpretations and to enhance the readability of the manuscript. We are convinced that this review is well placed in ACP. While most of the works we cited were published in other journals, we believe it is important to bring this information to atmospheric chemists and physicists who study ions in the atmosphere and make use of the ion-ion recombination coefficient; especially because, apart from the value of $1.6 \cdot 10^{-6}$ cm$^3$ s$^{-1}$ at standard conditions, the extensive field of fundamental research on this topic is unknown to many atmospheric scientists. Therefore, we tailored the scope of our review manuscript to compare and evaluate the existing theories and parameterisations for conditions of the troposphere and stratosphere. In fact, it was our initial motivation to review the theories on ion-ion recombination because we needed accurate values of $\alpha$ in different altitudes to be able to analyse airborne field data (see Zauner-Wieczorek et al. (2022)). We would like to make the existing knowledge available to the atmospheric science community.

References

Hickman, A. P.: Approximate scaling formula for ion–ion mutual neutralization rates, The Journal of Chemical Physics, 70, 4872–4878, https://doi.org/10.1063/1.437364, 1979.

Johnsen, R., Shun'ko, E. V., Gougousi, T., and Golde, M. F.: Langmuir-probe measurements in flowing-afterglow plasmas, Physical review. E, Statistical, nonlinear, and soft matter physics, 50, 3994–4004, https://doi.org/10.1103/PhysRevE.50.3994, 1994.

Loeb, L. B.: Basic Processes of Gaseous Electronics, Second edition, revised, University of California Press and Cambridge University Press, Berkeley, Los Angeles and London, 1960.

López-Yglesias, X. and Flagan, R. C.: Ion–Aerosol Flux Coefficients and the Steady-State Charge Distribution of Aerosols in a Bipolar Ion Environment, Aerosol Science and Technology, 47, 688–704, https://doi.org/10.1080/02786826.2013.783684, 2013.

Miller, T. M.: Parametrization of ion–ion mutual neutralization rate coefficients, The Journal of Chemical Physics, 72, 4659–4660, https://doi.org/10.1063/1.439711, 1980.

Miller, T. M., Shuman, N. S., and Viggiano, A. A.: Behavior of rate coefficients for ion-ion mutual neutralization, 300-550 K, The Journal of Chemical Physics, 136, 204306, https://doi.org/10.1063/1.4720499, 2012.

Shuman, N. S., Wiens, J. P., Miller, T. M., and Viggiano, A. A.: Kinetics of ion-ion mutual neutralization: halide anions with polyatomic cations, The Journal of Chemical Physics, 140, 224309, https://doi.org/10.1063/1.4879780, 2014a.

Shuman, N. S., Miller, T. M., Johnsen, R., and Viggiano, A. A.: Mutual neutralization of atomic rare-gas cations (Ne(+), Ar(+), Kr(+), Xe(+)) with atomic halide anions (Cl(-), Br(-), I(-)), The Journal of Chemical Physics, 140, 44304, https://doi.org/10.1063/1.4862151, 2014b.

Shuman, N. S., Miller, T. M., Hazari, N., Luzik, E. D., and Viggiano, A. A.: Kinetics following addition of sulfur fluorides to a weakly ionized plasma from 300 to 500 K: rate constants and product determinations for ion-ion mutual neutralization and thermal

electron attachment to SF5, SF3, and SF2, The Journal of Chemical Physics, 133, 234304, https://doi.org/10.1063/1.3520150, 2010.

Smith, D. and Adams, N. G.: Ionic recombination in the stratosphere, Geophys. Res. Lett., 9, 1085–1087, https://doi.org/10.1029/GL009i009p01085, 1982.

Smith, D. and Church, M. J.: Ion-ion recombination rates in the earth's atmosphere, Planetary and Space Science, 25, 433–439, https://doi.org/10.1016/0032-0633(77)90075-7, 1977.

Weaver, A., Arnold, D. W., Bradforth, S. E., and Neumark, D. M.: Examination of the 2A'2 and 2E" states of NO3 by ultraviolet photoelectron spectroscopy of NO3−, The Journal of Chemical Physics, 94, 1740–1751, https://doi.org/10.1063/1.459947, 1991.

Wiens, J. P., Shuman, N. S., and Viggiano, A. A.: Dissociative recombination and mutual neutralization of heavier molecular ions: C10H8(+), WF5(+), and C(n)F(m)(+), The Journal of Chemical Physics, 142, 114304, https://doi.org/10.1063/1.4913829, 2015.

Zauner-Wieczorek, M., Heinritzi, M., Granzin, M., Keber, T., Kürten, A., Kaiser, K., Schneider, J., and Curtius, J.: Mass spectrometric measurements of ambient ions and estimation of gaseous sulfuric acid in the free troposphere and lowermost stratosphere during the CAFE-EU/BLUESKY campaign, Atmos. Chem. Phys. Discuss., https://doi.org/10.5194/acp-2022-238, 2022.

**Answers to Referee 3**

The manuscript authored by Zauner-Wieczorek et al. presents a good review of the historical theory development on ion-ion recombination under relevant conditions of the troposphere and lower stratosphere. The authors then made a simple sensitivity study on the limiting sphere theories and compared the different parameterisations of the theories to measurement data from a few laboratory and field as well as model results. The content of the work, especially the review part, is valuable. The comparison studies are a bit flimsy, without discussions on why some parametrisations worked poorly and there was no insights given for corrections or improvements. The clarity of the manuscript needs to be improved and the manuscript needs somewhat a major revision.

We would like to thank the referee for their feedback. Based on the feedback provided by this referee and the other two referees, we revised the manuscript to improve the readability, the structure, the notations and use of symbols, and the discussion of the inter-comparison.

Comments:

When talk about ion-ion recombination, could you please first of all provide the definition of ion? Do you also consider the recombination of charged aerosol particles as ion-ion recombination?

We added the definition of ion-ion recombination as opposed to ion-aerosol attachment in the introduction (Sect. 1):

p. 3, l. 42ff.: "While ion-ion recombination concerns the recombination of atomic or molecular ions or small molecular ion clusters, ion-aerosol attachment regards the interaction between an ion and a charged or neutral aerosol particle. Typically, aerosol particles are defined to have a size of 1 nm or bigger. As the ion-aerosol attachment coefficient depends on the size of the aerosol particle, the ion-ion recombination coefficient can be viewed as a special case of the former if the "aerosol particle" is considered to have ionic size and is singly charged."

You compared the different parameterisations on ion-ion recombination to a few laboratory, field and model results and demonstrated that some models clearly have poor performance but did not discuss the potential causes. Could you please elaborate on this and provide insights into how they may be corrected or further improved?

We rewrote the discussion part of the inter-comparison completely (now Sect. 7) and added a discussion of the possible causes for deviations or poor performances.

Based on the comparisons with laboratory, field and model data, you suggested Brasseur and Chatel 1983 over other parameterisations. Given the fact that it has the semi-empirical nature, it is expected to agree better with measurement data. The measurement data (whether it is Rosen&Hofmann, Gringel et al., Morita or Franchin et al.) are based on probing air ion concentrations. Air conductivity is intrinsically dependent on ion concentration. Then the uncertainty from measurement loss inside the instrumentation or the system cannot be avoided. This was not discussed in the manuscript when making suggestions on the choice of theory.

We added more information on how the data by Gringel et al. (1978), Rosen and Hofmann (1981), and Morita (1983) were retrieved (see Sect. 4). Furthermore, we pointed out that these data sets can be subject to systematic error in the discussion part of the inter-comparison chapter (Sect. 7.1):

p. 19, l. 540ff.: "In Fig. 2 (a) to (d), the field measurements Gr78, RH81, and Mo83 are shown for a better comparability. Note that the data are inaccurate below 10 km. For RH81, there are two data sets for altitudes above 32 km: one is calculated based on Eq. (2), the other one is based on Eq. (37). One should bear in mind that these data sets, which were determined with similar methods, may also suffer from systematic errors such as losses inside the instrument that were not accounted for, however, these are the most reliable data from field measurements available to this day."

You did not recommend Tamadate 2020 due to its resulting in large deviation from measurement data. It seems however that the authors did not perform a MD simulation as described in Tamadate et al. 2020, instead the authors used the formula listed in Table 2 and referred that as Tamadate 2020. However, this functional form is merely Filippov's approach, which is similar to Fuchs model, as described in Tamadate 2020.

We rewrote the former chapter on limiting sphere theories (former Sect. 4), by separating and reorganising it into one chapter on ion-aerosol theories (now Sect. 5) and one on numerical simulations (now Sect. 6). The misconception of the work of Tamadate et al. (2020) was corrected and their work, i.e. a hybrid continuum-MD approach, is now described in Sect. 6. Fuchs's (1963) theory is explained in Sect. 5 and used for the comparison in Sect. 7.1. We renamed and relabelled the main text and the figure (now Fig. 2) accordingly.

I also find the manuscript was not very carefully prepared. The notations are especially confusing. For example, the mathematical symbol of prime should be used instead of ' (e.g. p6 L137). Also $d$ have several definitions through the manuscript, which is confusing. $v_+$ and $v_-$ were not defined where they appear first and definitions of $U_+$ and $U_-$ in eq 8 were missing. It is also unclear what is $x$ on p5 L128. A few different notations were used for the same property, e.g. $e$ and $e_T$ for collision probability, $d$ and $d_3$ for three-body trapping distance, etc. It is also sometimes difficult to distinguish between similar symbols like $a$ and $a$ and $M$ for molar mass and [M] for number density of air molecules. Please revise the manuscript carefully and drop off the repeated notions and use symbols that can be better distinguished.

We are thankful for the suggestions to enhance clarity in notation. We introduced the prime symbol instead of the apostrophe where applicable. We revised all radii and introduced a uniform notation as follows: $d$ is the three-body trapping distance; $\delta$ is the limiting sphere distance; we changed the collision radius from $a$ to $r_{coll}$. Indices further define $d$ and $\delta$ depending on the different theories. We introduced the symbol $\varepsilon$ to all collision probabilities, also using indices to indicate the specific definition of the different theories. For a better distinction, we changed the symbol of the ion mass from M to $m_{ion}$. We introduced $v_+$ and $v_-$ where they first appeared (now Eq. 4) and changed $U_+$ and $U_-$ in former Eq. 8 (now Eq. 9) to $v_+$ and $v_-$, which was a remnant of an older version of the draft. We added an explanatory sentence to define $x$: "In subsequent works, the ratio of the collision sphere radius and the mean free path of the ion, $2d_T \cdot \lambda_{ion}^{-1}$, is often denoted as $x$." We omitted repeated notions and only introduce the physical quantities in the main text where they first appear. The nomenclature in the end of the manuscript that contains all symbols and their explanations was revised accordingly.

p7 L160-161. It is confusing that you talk about 'collision probability becomes almost 0' and then 'collision is governed by the collision cross section'. Could you please elaborate what you mean here? How do you distinguish 'collision probability' and 'collision cross section'? To my understanding, the CCS is just a different way to quantify the probability of successful collisions.

Indeed, the ion-molecule collisions (leading to the dissipation of energy to enable the recombination) and the ion-ion collisions (i.e. the recombination itself) were not clearly distinguished in the former version of the manuscript. We, therefore, adapted the whole paragraph to make this distinction clearer. It now reads as follows:

p. 7, l. 169ff.: "Above atmospheric pressure, Langevin theory is applied. Loeb subclassifies it, firstly, to the range of 20 to 100 atm where there is no diffusional approach of the ions towards each other because they are already within the Coulomb attractive radius $d_T$ and, secondly, to the range of 2 to 20 atm (called Langevin-Harper theory), where the initial distance of the ions $r_0$ is greater than $d_T$ and so they first have to diffuse towards each other. The subsequent collision inside $d_T$ is almost certain because of the high pressure. For the pressure range of 0.01 to 1013 hPa, i.e. for the lower and middle atmosphere, Thomson theory is applicable. Here, the initial distance of the ions is greater than $d_T$ and the mean free path $\lambda_{ion}$, therefore a random diffusive approach is necessary. Within $d_T$, the collision probability $\varepsilon_T$ is less than 1. Below 0.01 hPa, i.e. in the ionosphere, the collision probability becomes almost 0 and, thus, the collision is then governed by the collision cross section. For super-atmospheric pressures (i.e. in the Langevin regime), $\alpha$ is dependent on $p^{-1}$ and proportional to $T$. In the regime where the Thomson theory should be applicable (i.e. from 0.01 to 1013 hPa), $\alpha$ is dependent $T^{-1.5}$, The pressure dependence of $\alpha$ is different in various Thomsonian theories; while Thomson (1924) stated a proportional dependence (see Eq. (4)), it varies in the parameterisations of Gardner (1938), Israël (1957), and Loeb (1960) (see Eq. (10), (14), (15), respectively, with Eq. (11) to (13)): for approximately 500 to 1000 hPa, $\alpha$ is dependent on $p^{0.5}$ and below 500 hPa, it approaches $p^1$. In the cross section regime (i.e. < 0.01 hPa), $\alpha$ is independent of the pressure and dependent on $T^{-0.5}$."

After Tamadate et al. (2020), the collision probability is the ratio of recombined ions over all ions that entered the limiting sphere; whereas the collision cross section is defined by $\pi (r_+ + r_-)^2$ and is, thus, a measure for the probability for binary recombination.

p7 L177. normal value? what is not normal?

For more clarification, we elaborated more on what the cited authors referred to as "normal" (as compared to "standard") conditions or values in the manuscript. The respective paragraphs now read as followed:

p. 4, l. 82f.: " Lenz (1932) reported $(1.7 \pm 0.1) \cdot 10^{-6}$ cm$^3$ s$^{-1}$ for the conditions of 291.15 K and 1013 hPa."

p. 8, l. 189ff.: "In the derivation of the formula he [Israël, 1957] used the value of $1.6 \cdot 10^{-6}$ cm$^3$ s$^{-1}$ for $\alpha$ for "normal conditions", however, he neither included a reference for this nor specified the normal conditions. These are probably 273.15 K and 1013.25 hPa."

P13 L367. what do you mean by 'ion current'?

In Hoppel and Frick's (1986) derivation of their Equation 4, given in our manuscript in former Eq. 32 (now omitted), they state: "Diffusion-mobility theories calculate the current of ions, $I_i$, to a particle from the steady-state diffusion-mobility equation. [...] The inner boundary conditions distinguish the various theories and match the diffusion-mobility flux outside some limiting sphere to the microscopic flux inside. The limiting sphere is concentric with the particle and has a radius $\delta'$ about one mean free path larger than the particle radius. [...] Matching the two fluxes at $\delta'$ yields [Equation 4]." In the course of revising the manuscript and the description of Hoppel and Frick's (1986) theory, the manuscript does not mention the ion current anymore.

p18 L472. what do you mean by 'trapping sphere'? Is it different from limiting sphere?

We added a more detailed explanation of the trapping sphere and the limiting sphere in the introduction of the new Sect. 5 (Application of ion-aerosol theories):

p. 14, l. 379ff.: "While in many ion-ion recombination theories, the concept of the three-body collision radius, or trapping radius, $d$, can be found, many ion-aerosol theories additionally use the concept of the limiting sphere, $\delta$. The limiting sphere and its radius are defined slightly differently depending on the theory. With Fuchs (1963), it is defined as a concentric sphere around the particle with the radius $\delta_F = r_p + \lambda'$, where $r_p$ is the particle radius and $\lambda'$ is "the mean distance from the surface of the particle at which the ions collide for the last time with gas molecules before striking this surface" (Fuchs, 1963). Notably, $\lambda'$ is not equal to the mean free path of one ion, $\lambda_{ion}$, or the ion-ion mean free path, $\lambda$. With Hoppel and Frick (1986), it is defined as the sum of the ion-aerosol three-body trapping sphere and the ion-ion mean free path (see Eq. (44)). Transferred to the ion-ion recombination, the limiting sphere can be defined as the sum of the ion-ion aerosol three-body trapping and one mean free path (see Eq. (45)), as depicted in Fig. 1."

p24 L587. Ta20 yields α values which are one order of magnitude too low (2.7 · 10–6 cm3 s–1 at ground level). Is it true? 2.7e-6 cm3s-1 does not seem too low.

We thank the referee for pointing out this typo. It should read $2.7 \cdot 10^{-5}$ cm$^3$ s$^{-1}$, but the main text of the comparison chapter was revised completely for the updated manuscript.

Fig.1 caption. please consider using open circle instead of white point.

We appreciate this suggestion, but the white point/circle is not open, but filled white. In the further text, referring to the "open-circle ion" would be more cumbersome than referring to the "white ion", so that we decided to keep the initial wording.

Fig.3c The color for Tamadate et al. 2020 in legend is different from that in the plot.

We decided to omit the legends and added labels within the plots to enhance comprehensibility.

Table 1. please define the symbols in the caption. what is $r_0$?

We added the definition of symbols in the table caption, including $r_0$, the initial distance of the two ions. We changed $d$ to $d_T$ (Coulombic radius after Thomson) to distinguish it from other radii discussed in the manuscript.

I also suggest that you consider restructuring some parts of the text. I find organisation of section 2 in the current manuscript does not render a smooth textflow, especially concerning the definition of $d$. Because $d$ appears earlier in the text already but its definition comes quite late. Also in section 4, there is a sudden jump to ion-aerosol attachment without preparing the readers with the purpose.

We followed the suggestion of the referee and reorganised parts of the manuscript. Now, the three-body trapping distance $d$ is introduced and explained at its first appearance in Sect. 2. We added a chapter (Sect. 8) to determine $d$ numerically by solving Natanson's (1959) equation for $d$, using the field data of $\alpha$ by Gringel et al. (1978), Rosen and Hofmann (1981), and Morita (1983) as input parameters. The short chapter on the sensitivity study of Hoppel and Frick's and Natanson's $d$ (former Sect. 5.1) was replaced by it. Furthermore, we reorganised the chapter on ion-aerosol theories (now Sect. 5), explaining the motivation more clearly why to add this chapter.

References

Fuchs, N. A.: On the stationary charge distribution on aerosol particles in a bipolar ionic atmosphere, Geofisica pura e applicata, 56, 185–193, https://doi.org/10.1007/BF01993343, 1963.

Gardner, M. E.: The Recombination of Ions in Pure Oxygen as a Function of Pressure and Temperature, Physical Review, 53, 75–83, https://doi.org/10.1103/PhysRev.53.75, 1938.

Gringel, W., Käselau, K. H., and Mühleisen, R.: Recombination rates of small ions and their attachment to aerosol particles, Pure and Applied Geophysics, 116, 1101–1113, https://doi.org/10.1007/BF00874674, 1978.

Hoppel, W. A. and Frick, G. M.: Ion—Aerosol Attachment Coefficients and the Steady-State Charge Distribution on Aerosols in a Bipolar Ion Environment, Aerosol Science and Technology, 5, 1–21, https://doi.org/10.1080/02786828608959073, 1986.

Israël, H.: Atmosphärische Elektrizität: Teil 1. Grundlagen, Leitfähigkeit, Ionen, Akademische Verlagsgesellschaft Geest & Portig K.G., Leipzig, 1957.

Lenz, E.: Die Wiedervereinigung von Ionen in Luft bei niederen Drucken, Zeitschrift für Physik (Zschr. f. Physik), 76, 660–678, https://doi.org/10.1007/BF01341939, 1932.

Loeb, L. B.: Basic Processes of Gaseous Electronics, Second edition, revised, University of California Press and Cambridge University Press, Berkeley, Los Angeles and London, 1960.

Morita, Y.: Recent measurements of electrical conductivity and ion pair production rate, and the ion-ion recombination coefficient derived from them in the lower stratosphere, Journal of Geomagnetism and Geoelectricity, 35, 29–38, https://doi.org/10.5636/jgg.35.29, 1983.

Natanson, G. L.: The Theory of Volume Recombination of Ions, Journal of Technical Physics, 4, 1263–1269, 1959.

Rosen, J. M. and Hofmann, D. J.: Balloon-borne measurements of electrical conductivity, mobility, and the recombination coefficient, Journal of Geophysical Research: Oceans, 86, 7406–7410, https://doi.org/10.1029/JC086iC08p07406, 1981.

Tamadate, T., Higashi, H., Seto, T., and Hogan, C. J., J.: Calculation of the ion-ion recombination rate coefficient via a hybrid continuum-molecular dynamics approach, The Journal of Chemical Physics, 152, 94306, https://doi.org/10.1063/1.5144772, 2020.

Thomson, J. J.: XXIX. Recombination of gaseous ions, the chemical combination of gases, and monomolecular reactions, The London, Edinburgh, and Dublin Philosophical Magazine and Journal of Science, 47, 337–378, https://doi.org/10.1080/14786442408634372, 1924.